# Brain-optimized extraction of complex sound features that drive continuous auditory perception

Julia Berezutskaya[1,2]*, Zachary V. Freudenburg[1], Umut Güçlü[2], Marcel A. J. van Gerven[2], Nick F. Ramsey[1]

**1** Department of Neurology and Neurosurgery, Brain Center, University Medical Center Utrecht, Utrecht, The Netherlands, **2** Donders Institute for Brain, Cognition and Behaviour, Radboud University, Nijmegen, The Netherlands

* y.berezutskaya-2@umcutrecht.nl

**Data Availability Statement:** The code containing the BO-NN architecture and the details of training is available at https://github.com/Immiora/bonn_auditory_perception_ecog. Neural data cannot be

## Abstract

Understanding how the human brain processes auditory input remains a challenge. Traditionally, a distinction between lower- and higher-level sound features is made, but their definition depends on a specific theoretical framework and might not match the neural representation of sound. Here, we postulate that constructing a data-driven neural model of auditory perception, with a minimum of theoretical assumptions about the relevant sound features, could provide an alternative approach and possibly a better match to the neural responses. We collected electrocorticography recordings from six patients who watched a long-duration feature film. The raw movie soundtrack was used to train an artificial neural network model to predict the associated neural responses. The model achieved high prediction accuracy and generalized well to a second dataset, where new participants watched a different film. The extracted bottom-up features captured acoustic properties that were specific to the type of sound and were associated with various response latency profiles and distinct cortical distributions. Specifically, several features encoded speech-related acoustic properties with some features exhibiting shorter latency profiles (associated with responses in posterior perisylvian cortex) and others exhibiting longer latency profiles (associated with responses in anterior perisylvian cortex). Our results support and extend the current view on speech perception by demonstrating the presence of temporal hierarchies in the perisylvian cortex and involvement of cortical sites outside of this region during audiovisual speech perception.

## Author summary

A lot remains unknown regarding how the human brain processes sound in a naturalistic setting, for example when talking to a friend or watching a movie. Many theoretical frameworks have been developed in attempt to explain this process, yet we still lack the comprehensive understanding of the brain mechanisms that support continuous auditory processing. Here we present a new type of framework where we seek to explain the brain

shared publicly as part of the current submission due to confidentiality restrictions on sharing patient specific information. Neural data can be made available from the University Medical Utrecht center for researchers who meet the criteria for access to confidential data. Please contact Mariska J. van Steensel at M.J.vanSteensel@umcutrecht.nl or Annemiek Elbertse at a.elbertse-schouten@umcutrecht.nl. In addition, we are currently writing a dataset paper, along with which we will released the anonymized neural data used in the present study in the open database (www.openneuro.org).

**Funding:** This research was funded by the ERC-Advanced 'iConnect' project (grant ADV 320708), and the 'Language in interaction' project (NWO Gravitation grant 024.001.006). The funders had no role in study design, data collection and analysis, decision to publish, or preparation of the manuscript.

**Competing interests:** The authors have declared that no competing interests exist.

responses to sound by considering few theoretical assumptions and instead learn about the brain mechanisms of auditory processing with a 'data-driven' approach. Our approach is based on applying a deep artificial neural network directly to predicting the brain responses evoked by a soundtrack of a movie. We show that our framework provides good prediction accuracy of the observed neural activity and performs well on novel brain and audio data. In addition, we show that our model learns interpretable auditory features that link well to the observed neural dynamics particularly during speech perception. This framework can easily be applied to external audio and brain data and is therefore unique in its potential to address various questions about auditory perception in a completely data-driven way.

## Introduction

Our understanding of how the human brain processes auditory input remains incomplete. In general, our aim is to identify the features that different cortical regions extract from the incoming sound signal, and to understand how they are transformed into high-level representations specific to sound type (speech, music, noise, etc.).

Low-level acoustic sound properties, captured as audio envelope, spectrogram representation, gammatone-filtered representation [1], spectrotemporal modulations [2] and related measures, have proven quite effective in supporting development of neural encoding models of both synthetic and natural sound perception [3–7]. Addressing higher-level features has also been attempted in neural encoding models of sound processing [8,9], but higher levels of auditory processing are generally more difficult to model because their characteristics (e.g. in speech or music) remain a topic of theoretical investigation. Higher-level features typically require some form of interpretation and labelling that is based on theoretical constructs and therefore may not match cortical representations. Moreover, little is known about the mechanisms underlying the transition from lower- to higher-level auditory processing, leaving these levels of explanation disconnected.

A possible alternative to using predetermined stimulus features to investigate sound processing is to construct a hierarchical model of auditory processing that can learn the optimal multi-level auditory features from the data (bottom-up), with a minimal set of theoretical assumptions about the higher-level features. Recent advances in deep learning have shown a possibility of construction of such models in the field of object recognition, text processing and sound generation. Recent work in computational neuroscience indicates that the multi-level features captured in artificial neural networks (ANNs) relate to the hierarchy of cortical representations in visual perception [10–12], music [13] and language [14] processing.

There is a long line of research showing evidence of the hierarchical organization of information expressed in both human behavior [15,16] and the principles of neural processing in perception [17–20], action [21,22] and memory [23]. It is possible that the hierarchical structures in the neural responses themselves can be used to directly drive the construction of the encoding model of auditory processing.

Thus, in this study, we sought to develop an encoding model of auditory processing while injecting as little top-down theoretical assumptions or constraints as possible. We modeled the neural responses to sound directly by training a deep ANN on the raw audio input. We assumed that the mechanism of auditory input processing generalizes across individuals, and that models derived from one experimental setting should perform equally well in another.

We thus obtained a model from one dataset and applied it to data from new participants perceiving a different set of stimuli.

To this end, we collected neural responses to a full-length feature film from six patients implanted with intracranial electrodes for diagnostic purposes. A deep ANN was trained on the raw audio soundtrack to predict the associated brain responses in all covered regions directly through extraction of multi-layer representations of sound. The obtained model was then applied to data from new participants watching a different movie. Our results indicate a hierarchical processing structure in the brain during auditory perception that is generic across individuals and experimental material (different films and different participants). Moreover, the sound features extracted by the data-driven model captured acoustic properties specific to different types of sound. Crucially, without relying on any theoretical constraints the model learned to incorporate one of the most important characteristics of speech–its hierarchical temporal organization.

## Results

The goal of the present study was to construct an auditory perception neural model by predicting the cortical brain responses to the auditory input directly and in a bottom-up manner, injecting as little theoretical constraints to our encoding model as possible. For this, we collected electrocorticography (ECoG) data from six patients who watched a long-duration feature film (Movie I, 78 minutes). Next, a deep ANN was trained on the raw soundtrack of the movie to predict the associated ECoG responses in the high frequency band (HFB, 60–95 Hz) [24]. We then confirmed that our brain-optimized ANN (BO-NN, **Fig 1A**) model could be successfully applied to a dataset of different participants watching a different audiovisual film (Movie II). The BO-NN model extracted features that more closely matched the neural responses in the perisylvian cortex compared to commonly used theory-driven features (spectrotemporal modulation features). Finally, identification and visualization of the key features of the BO-NN model showed the presence of features specific to certain sound types, which were also associated with distinct cortical locations, acoustic sound properties and latencies with respect to sound onset.

### Architecture of the BO-NN model

There are many architectural choices associated with constructing an ANN. Optimization of ANN parameters is possible but computationally costly and, importantly, requires large amounts of data. In our case of only 78 minutes of continuous brain recordings, it may not be ideal to expend these valuable data on a thorough optimization. At the same time, it is conceptually interesting how different design solutions affect performance. Here, we took a midway approach and explored relations between several architectural choices and the model performance without running a full optimization procedure. In particular, we explored the effects of the computational mechanisms in the nodes (convolutional, CNN, recurrent, RNN and a combination of these mechanisms, RCNN), the type of the input audio data (time-domain, spectrogram or both), the depth of the ANN, and the size of the temporal window in the input data necessary for a prediction of the associated brain responses.

Each ANN was trained and tested on the dataset of Movie I. The Spearman correlation between predicted and observed HFB responses per electrode was used as the model performance metric. The performance accuracy was cross-validated over ten test folds by computing the Spearman correlation in a held-out test set per fold (10% of the movie: ~7.8 min, 23,620 time points) and then averaging over all folds. The difference in model performance was assessed with non-parametric Wilcoxon signed-rank tests (*Z*-statistic reported).

## a

### Architecture of brain-optimized (artificial) neural network (BO-NN)

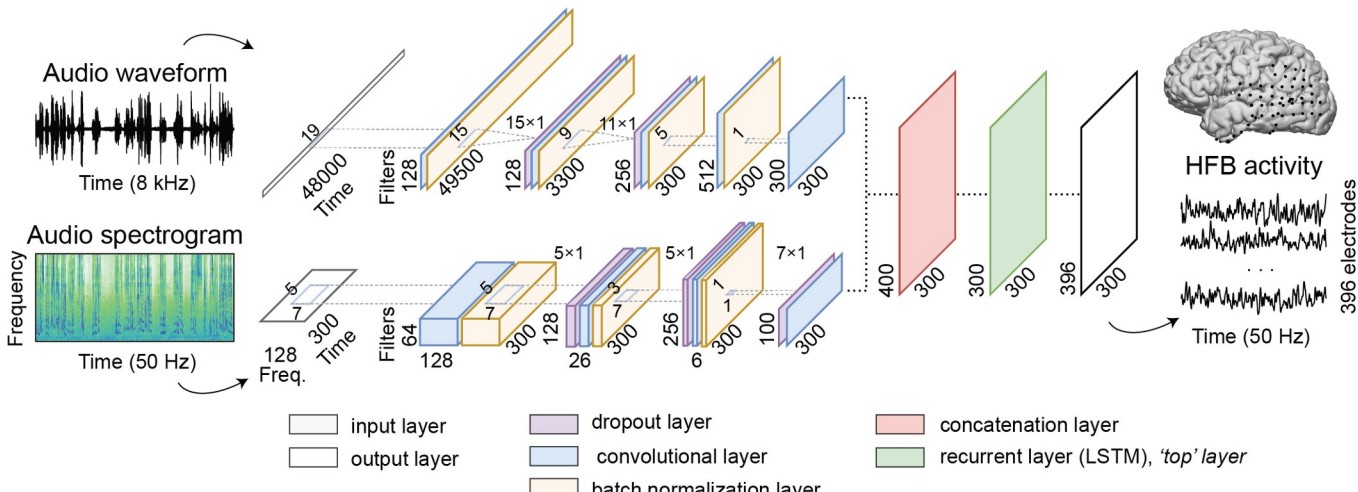

## b

### Testing various architectural choices

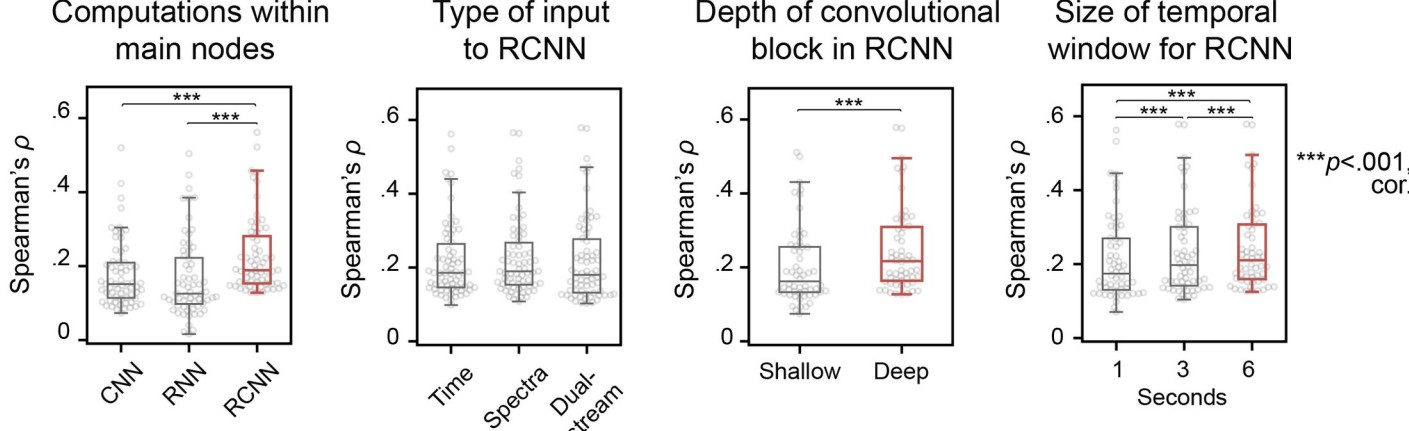

**Fig 1. Architecture of the BO-NN model. (a)** The BO-NN model received two inputs: a 1D time-domain audio signal at 8 kHz (audio waveform) and a 2D time-frequency signal at 50 Hz (audio spectrogram). Six-second audio chunks were input to the model at once (48000 and 300 time points, respectively). Two convolutional streams were trained to process each input separately. Then, both representations were concatenated and passed to the recurrent layer. The output was a prediction of the HFB neural signal at 50 Hz (300 time points over 396 electrodes). **(b)** Results of testing various ANN architectures. Dots represent cross-validated model performance for individual electrodes. Boxes outline the 25th and 75th percentiles of the model performance over electrodes with a significant fit, caps show 5th and 95th percentiles. A solid line within each box is the median. Red box outlines the model with a significantly better cross-validated performance. Four comparisons were performed: (1) computations within the ANN nodes (convolution, CNN, recurrence, RNN and a combination of both, RCNN); (2) the type of the audio input (a waveform, a spectrogram and a combination of both); (3) depth of the convolutional block of the ANN (one layer vs multiple layers); and (4) the size of the temporal window in the audio input (1-second, 3-second and 6-second). For each comparison a set of ANNs with relevant architectures were constructed and trained on the data of Movie I. In general, wherever possible we matched the number of parameters across the different ANNs. Spearman correlation ($\rho$) between the predicted and observed HFB responses was used as the model performance metric. The performance of each model was cross-validated over ten test folds. A strict parametric threshold $p<1\times10^{-20}$ on the $t$-transformed $\rho$ values was used to estimate significance of the fit per electrode. Other thresholds were tested as well ($1\times10^{-10}$, $1\times10^{-50}$, $1\times10^{-80}$ and $1\times10^{-100}$) and yielded the same results in all ANN comparisons. The electrodes with a significant ANN fit (reported here at $p<1\times10^{-20}$) in any of the models were selected for each comparison (for example when comparing the computations in the main nodes we selected all electrodes with a significant fit in any of the three models: CNN, RNN or RCNN). Non-parametric Wilcoxon signed-rank tests were applied to compare the cross-validated model performance over the ANNs in each comparison. The reported results were significant at $p<.001$ (***), Bonferroni corrected for the number of the ANN models and the number of architecture comparisons.

First, we observed that a model that combined convolutional and recurrent nodes (RCNN) provided a better fit ($Z_{RCNN-CNN}$ = 11.19 and $Z_{RCNN-RNN}$ = 11.2, both at $p \ll .001$, Bonferroni corrected for the number of models) for a larger number of electrodes (64 in RCNN, 45 in CNN and 31 in RNN with fit significant at $p < 1 \times 10^{-20}$, Bonferroni corrected for number of electrodes) compared to the models that only used one type of computation, be it convolutional or recurrent (**Fig 1B**). Second, the RCNN model performed well regardless of the type of input data (time-domain, spectrogram or both): $Z_{both-time}$ = 2.22, $p$ = .03 and $Z_{both-spectra}$ = 0.7, $p$ = .05. Third, an RCNN with a deeper convolutional component (four convolutional layers) performed better compared to a more shallow RCNN (one convolutional layer): $Z_{deep-shallow}$ = 10.61, $p \ll .001$. Finally, feeding in longer audio fragments at a time (3- or 6-second long) led to better performance compared to feeding in shorter fragments (1-second long): $Z_{6sec-1sec}$ = 11.07 and $Z_{6sec-3sec}$ = 8.3, both at $p \ll .001$, Bonferroni corrected for the number of models.

These comparisons confirm that the architectural choices can affect the model performance given the limited amount of data. They also suggest that the hierarchical integration of temporal and spectral audio properties (multi-layer convolutions) over multi-second fragments of sound data (3- to 6-second long) combined with a complex analysis of the previous temporal history (recurrence) give rise to representations that match well with the brain responses to sound.

## BO-NN model performance (6 participants, Movie I)

Taking into account the results of the architecture comparisons, we selected the deep ANN model with two separate streams of convolutional layers (temporal and spectral representations of audio input), the outputs of which were combined and passed to a recurrent module (the RCNN architecture). The model looked at six seconds of input audio data at once. We refer to this final configuration as the brain-optimized neural network (BO-NN) model and explore below further details of its performance.

The highest cross-validated prediction accuracy (Spearman correlation between predicted and observed HFB responses over ~7.8 min of Movie I, 23,620 time points) was achieved for electrodes in superior temporal gyrus (STG), reaching $\rho_{max} \approx .5$, compared to inferior frontal gyrus (IFG), precentral, postcentral and middle temporal gyrus (MTG) areas ($\rho_{max} \approx .3$, **Fig 2C**). The BO-NN model fitted 83% of all STG electrodes compared to 46%, 23%, 17% and 37% of IFG, precentral, postcentral and MTG electrodes, respectively (at $p < 1 \times 10^{-20}$, Bonferroni corrected for number of electrodes). There was no coverage of the Heschl's gyrus in our electrode set. The prediction accuracy did not differ between various types of sound present in the soundtrack of Movie I (speech, music, noise and ambient sounds, **Fig 2A**) when we analyzed them separately (**Fig 2B**). We did not observe selectivity to a specific sound type in any of the fitted electrodes (based on Wilcoxon's signed-rank tests on prediction accuracy across sound types, also see **Fig 2B and 2D**). This result could be a consequence of using a continuous sound stimulus with an unbalanced distribution over different types of sound (**Fig 2A**).

In subjects with substantial dorsal lateral occipitotemporal complex (LOTC) coverage (S1, S4, S5, S6) many electrodes were also predicted significantly well, potentially indicating presence of audiovisual interactions in naturalistic stimuli.

## BO-NN model performance with unseen data (29 new participants, Movie II)

Next, we sought to test how well the BO-NN model could generalize to unseen audio and brain data. The performance of the BO-NN model was therefore validated on a separate movie

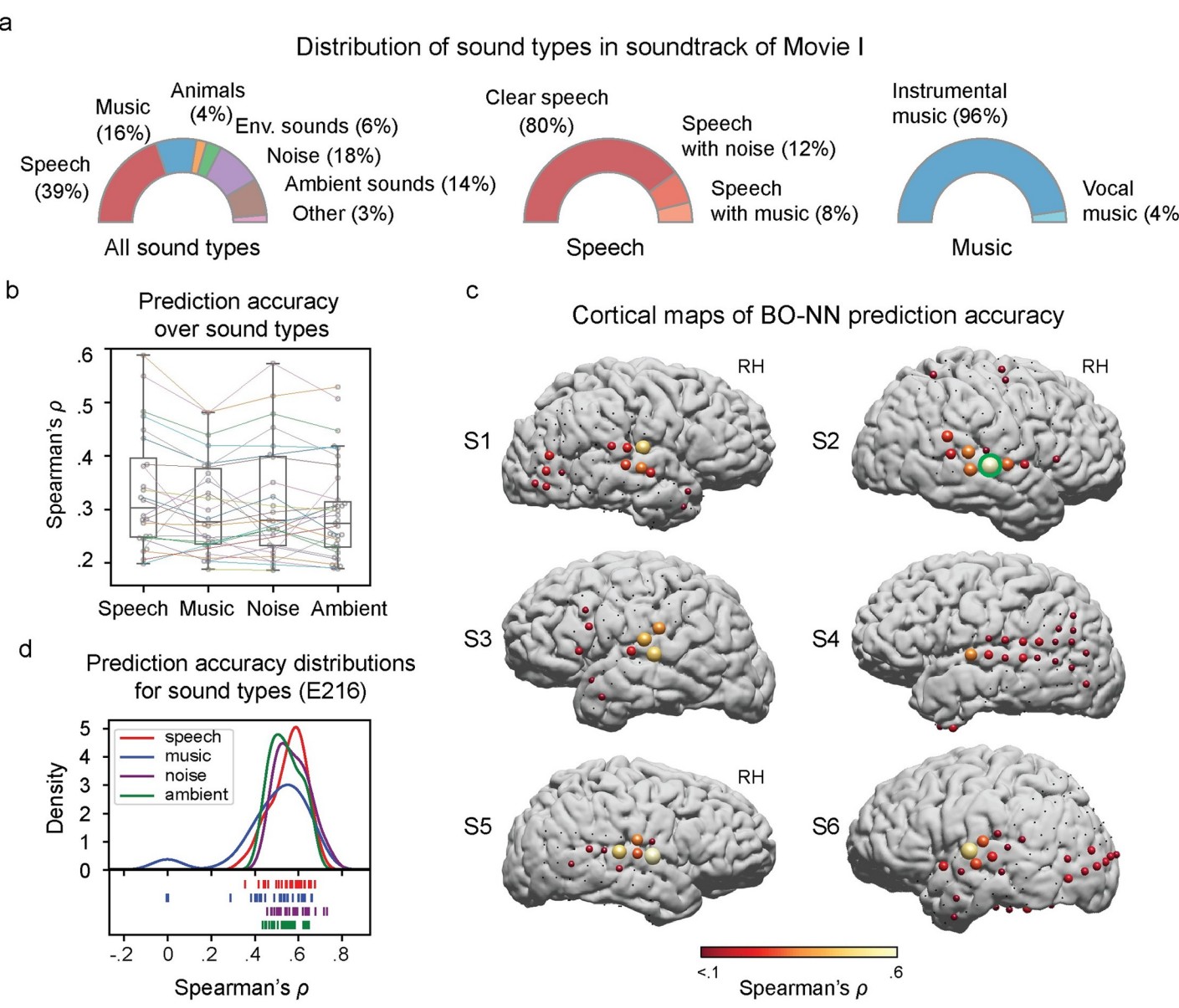

**Fig 2. BO-NN model performance with Movie I (6 participants). (a)** Overview of the training data (Movie I). The distributions of sound types were made based on the manual annotation of the soundtrack. **(b)** Cross-validated BO-NN model performance (over ten folds) estimated as the Spearman correlation ($\rho$) between the predicted and observed HFB responses on test set of Movie I. Spearman's $\rho$ is shown per type of sound with a sufficient amount of data (at least 10% of the soundtrack): speech, noise, music and ambient sounds. In each test fold all fragments of the same sound type were concatenated, then the Spearman's $\rho$ was calculated per each sound type and averaged over ten cross-validation folds. Each dot is an ECoG electrode with a significant fit at $p < 1 \times 10^{-20}$, Bonferroni corrected for the number of electrodes and folds. Boxes show 25th and 75th percentiles, caps show 5th and 95th percentiles. A solid line within each box shows the median. The colored lines connect the model performance for the same electrodes across the sound types. The colored lines are less visible for smaller $\rho$ values, however it is clear from the plot that five electrodes ($\rho \geq .4$) were associated with comparable prediction accuracy across different sound types. **(c)** Projection of electrodes with the significant cross-validated model performance (Spearman's $\rho$ for ~7.8 min of data) on individual cortical surfaces. Spearman's $\rho$ values significant at $p < 1 \times 10^{-20}$, Bonferroni corrected for the number of electrodes and folds are shown. The size of each colored electrode is proportional to the $\rho$ value that is additionally color-coded. Small black dots show electrodes that did not obtain a significant BO-NN fit and are displayed for the coverage reference. **(d)** Distribution of the prediction accuracy over different sound types for an example electrode E216 (outlined in green on the cortical map in **c**). First, we concatenated all ten test folds into a full soundtrack of predicted HFB responses. Then, per sound type we sampled 30 fragments, each 30-second long, at random starting points throughout the soundtrack and correlated the predicted and observed HFB responses. The rug plots show $\rho$ values per individual 30-second fragment and per sound type. The probability density plots outline the distribution of the $\rho$ values per sound type. This plot is another illustration that the best fitted electrodes were associated with high prediction accuracies regardless of the type of sound.

## BO-NN performance in Movie II

### Speech

### Music

**Fig 3. BO-NN model performance with unseen data (29 new participants, Movie II).** Cross-validated model performance (over six folds) estimated as Spearman's $\rho$ values (similar to Movie I) is projected onto the corresponding electrode locations on the average MNI cortical surface. The displayed results were obtained with the features from the top (recurrent) layer of the BO-NN model. Individual electrode locations were normalized to the MNI space using patient-specific affine transformation matrices obtained with SPM8. Spearman's $\rho$ values significant at $p < .001$, Bonferroni corrected for the number of electrodes, are shown (significance testing was based on a surrogate distribution of shifted data, see Methods for details). The BO-NN model performance is shown separately for speech and music test sets. Note the difference in the range of the Spearman's $\rho$ values between speech (up to .6) and music (up to .3). For the visualization purposes, a 2D Gaussian kernel (FWHM = 8 mm) was applied to the coordinate on the brain surface corresponding to the center of the electrode, so that the model performance score (Spearman's $\rho$ value) faded out from the center of the electrode toward its borders. Small black dots show electrodes that did not obtain a significant BO-NN fit and are displayed for the coverage reference.

watching dataset from 29 new ECoG patients (Movie II, 6.5 minutes with 13 interleaved 30-second blocks of speech and music). The soundtrack was passed through the pretrained model with all weights fixed, to obtain BO-NN activations. A regularized linear regression was trained per BO-NN layer to predict ECoG responses. Activations of the top layer of the model (recurrent layer) were best at predicting the neural responses compared to other layers (two-sided Wilcoxon signed-rank tests, 14 in total: $\bar{Z} = 14.55 \pm 1.81, p \ll .001$, Bonferroni corrected for number of layers).

The results of the BO-NN fit (top layer) during speech fragments of Movie II were comparable to the results obtained in Movie I in terms of prediction accuracy and cortical distribution (**Fig 3**). Good performance was obtained mostly in the perisylvian cortex for both Movie datasets (Heschl's gyrus was not covered in Movie II either). Prediction of music fragments was less accurate and the sound selectivity analysis showed that 83 electrodes in STG and IFG were fitted significantly better during speech compared to music fragments with no electrodes fitted better during music. This difference in prediction accuracy over sound types compared to the results in Movie I was not due to the difference in the music material across the Movies as the acoustic similarity of music across the movies ($\bar{s}_{music(I,II)} = 89 \pm .03$) was not significantly different from the acoustic similarity of speech ($\bar{s}_{speech(I,II)} = 91 \pm .02$), see Methods for details). Instead, the difference was possibly due to the attentional mechanisms arising from the experimental manipulation of the soundtrack in Movie II. Specifically, unlike the sound of Movie I (which was a full-length feature film), the sound of Movie II had a predictable structure of interleaved 30-second blocks of speech and music. The speech blocks contained dialogues relevant for the development of the story whereas music blocks did not. This could have triggered attentional shifts between the speech and music blocks that affected the HFB

responses to music. Of interest, the amplitude of the HFB responses to speech in the perisylvian cortex was on average higher compared to music (signed $\bar{r}^2 = .77$, see Methods for details).

Given that overall the BO-NN model obtained with Movie I data generalized well to Movie II data, and the latter contained considerably more electrodes (2218 compared to 396), all further analyses were carried out using data from Movie II. Moreover, using a model that was developed on an independent dataset with separate patients strengthens the generalizability of findings.

## Comparison of the data-driven BO-NN features with control feature sets

Next, we sought to understand how our BO-NN features in the top layer (recurrent layer of BO-NN) compared to alternative, control features sets using data of Movie II. On the one hand, we sought to compare them against a commonly-used theory-driven feature set, that has been shown to explain considerable variance in the neural responses [3,4,25]. On the other hand, we intended to check that BO-NN model benefitted from training on the neural responses, and did not offer a good fit simply due to its complex multi-layer architecture. Thus, we compared top-layer BO-NN features to theory-driven spectrotemporal modulation features (STMFs) [2], on the one hand, and the top layer of an ANN with the same architecture as the BO-NN model but with randomly initialized weights (Rand-NN) on the other. STMFs were extracted from the sound spectrogram using 2D convolutions and were organized along three dimensions: temporal and spectral sound modulations (TMs and SMs, respectively) and frequency. For a fair comparison, we ensured the optimal extraction of STMF features (see Methods for details). Random weights of Rand-NN were sampled from the zero-mean multivariate Gaussian distribution. Given the difference in fitting speech and music responses in Movie II, we carried out all analyses separately for speech and music.

**Sound representation in STG.** Data-driven (BO-NN) and theory-driven (STMFs) feature sets were compared to the neural responses in STG using a representational similarity analysis (RSA) [26,27]. RSA allows estimation of how similar the inner structure in one dataset is to the inner structure of other datasets (see Methods for details). Here, for speech, all three feature sets exhibited significant similarity with the representations in STG (one-sided Wilcoxon signed-rank tests: $\bar{Z} = 4.78, p < .001$, Bonferroni corrected for number of feature sets), however top-layer BO-NN representations were most similar to STG (two-sided Wilcoxon signed-rank tests: $Z_{BONN-STMF} = 3.98$ and $Z_{BONN-RANDNN} = 4.78$, both at $p<.001$, Bonferroni corrected for number of feature sets, **Fig 4A**, top panel). In music however, neither feature set showed better similarity with STG responses. This difference between speech and music conditions also suggests that BO-NN does not simply provide a better fit of the overall shape of the HFB responses (since it was trained to predict neural data whereas STMFs are a theory-driven acoustic model and Rand-NN was untrained). This result likely means that BO-NN captures something about the speech representation STMFs and Rand-NN do not.

**Predictions in higher-level speech regions.** Given the observed difference for STG, we further examined responses in higher-level speech processing regions. Three regularized linear regression models were trained to predict the neural responses to speech from BO-NN (top layer,1), STMFs (2) or Rand-NN (3) features. Compared to both STMFs and Rand-NN, top-layer BO-NN reached higher prediction accuracies and fitted more electrodes not only in STG (two-sided Wilcoxon signed-rank test: $Z_{BONN-STMF} = 4.34, p<.001$, +8 sign. els., $Z_{BONN-RANDNN} = 8.94, p<.001$, +49 sign. els., **Fig 4A**, bottom panel), but also in the higher-level

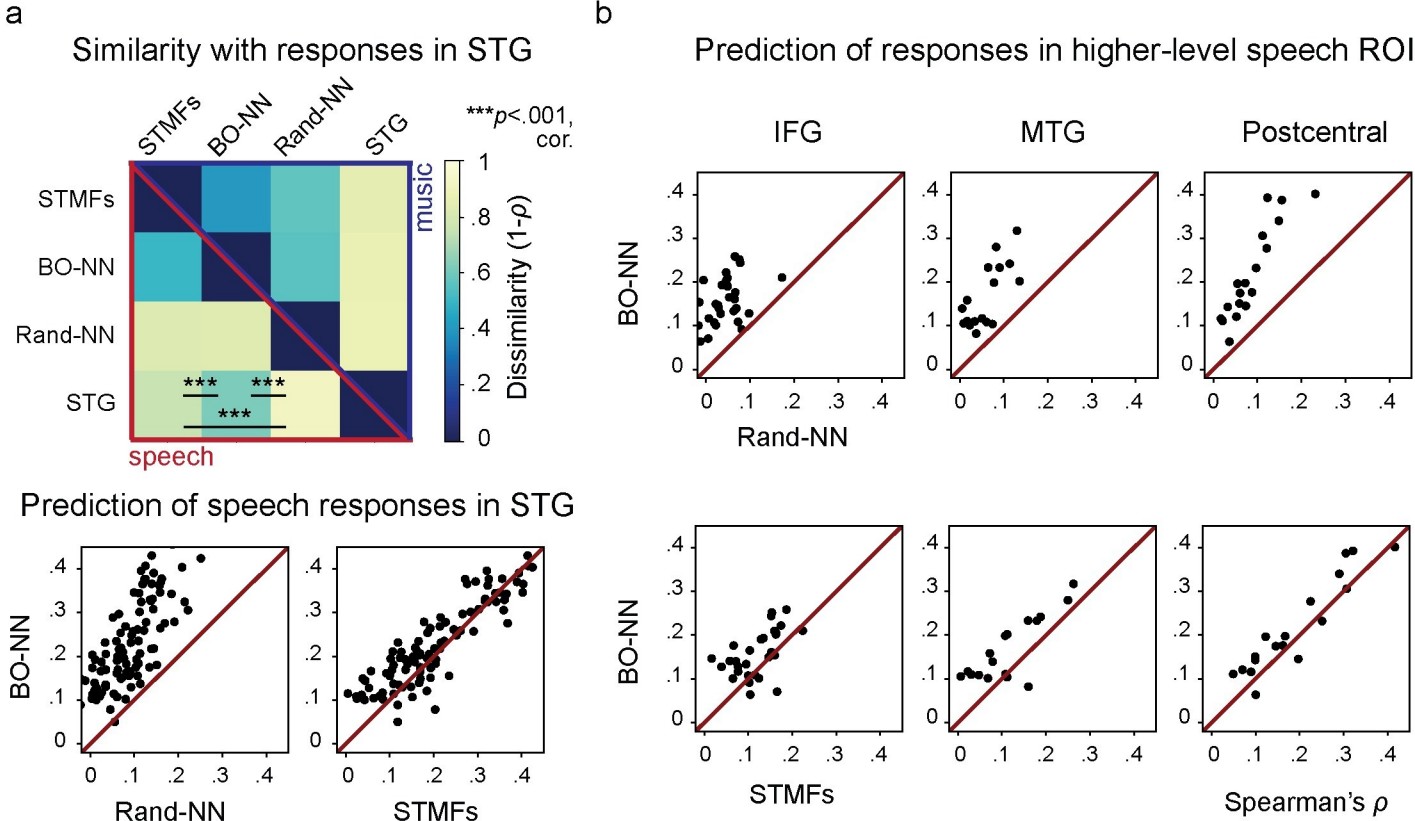

**Fig 4. Comparison of the data-driven BO-NN model trained on the neural responses with two control feature sets for Movie II.** First control feature set is the theory-driven spectrotemporal modulation feature (STMF) model. Second control feature set is the top layer of the ANN with the same architecture as the data-driven BO-NN but not optimized to fit the neural responses (Rand-NN). **(a)** Top panel displays second-order dissimilarity matrices showing how dissimilar the audio representations are between STG, the top layer of the data-driven BO-NN model, the theory-driven STMF model and the top layer of the non-optimized Rand-NN. The dissimilarity matrices are shown separately for speech and music data. The degree of similarity between the three audio representations with STG was tested using Wilcoxon signed-rank tests (see main text and Methods for details). The results were significant only for speech at $p<.001$, Bonferroni corrected for the number of models. Bottom panel shows the cross-validated prediction accuracy in speech (over six folds) achieved by the top layer of the BO-NN model per individual STG electrode. Scatter plots display the difference in prediction by the data-driven BO-NN model (top layer) from the prediction by the non-optimized Rand-NN model (top layer) and the theory-driven STMF model. Each dot is an ECoG electrode with significant Spearman's $\rho$ between the predicted and observed HFB responses on the test set of Movie II. Spearman's $\rho$ significant at $p<.001$, Bonferroni corrected for the number of cortical regions is shown. **(b)** Cross-validated prediction accuracy (over six folds) achieved by the top layer of the BO-NN model per individual electrode in higher-level speech ROIs: IFG, MTG and postcentral gyrus. The mode of display is analogous to the STG plots from **a**.

speech processing areas: IFG ($Z_{BONN-STMF} = 3.30, p<.001$, +4 sign. els., $Z_{BONN-RANDNN} = 4.46$, $p<.001$, +25 sign. els.), MTG ($Z_{BONN-STMF} = 3.01, p<.003$, +5 sign. els., $Z_{BONN-RANDNN} = 3.62$, $p<.001$, +13 sign. els.) and postcentral gyrus ($Z_{BONN-STMF} = 2.46, p<.01$, +3 sign. els., $Z_{BONN-RANDNN} = 3.72, p<.001$, +11 sign. els., **Fig 4B**). Importantly, we observed that the fit using only 300 BO-NN features was superior to the STMFs even when aggregating STMFs over multiple temporal delays within 500 ms from the audio onset (see Methods for more details). The improvement was significant for STG ($Z_{BONN-STMF\_DELAYS} = 3.93, p = 8\times10^{-5}$, +7 sign. els.) and MTG ($Z_{BONN-STMF\_DELAYS} = 2.15, p =.03$, +2 sign. els.). These results suggested that the BO-NN model captured features that better explained the neural responses to speech in STG as well as in higher-level speech processing areas compared to various control feature sets even when the features of the control feature sets were aggregated over multiple lags to account for the temporal delay of the neural response to sound.

## Visualization and interpretation of key BO-NN features

Having observed high prediction accuracy across datasets and subject groups, as well as the improvement over the state-of-the-art theory-driven features in fitting the neural responses to speech, we focused on visualization and interpretation of the BO-NN features. We only explored the top layer of the BO-NN model (recurrent layer, referred to as "top BO-NN" from now on) since these features were best at predicting the ECoG responses. To this end, we used feature visualization, clustering, projection of the associated cortical weight maps (β-weights of the linear fit of the neural responses from top BO-NN) and relation to the known sound properties to gain insight about each data-driven feature.

**Identification of key BO-NN features.**   To minimize the number of features to work with, we first identified the key features of top BO-NN using data-driven affinity propagation clustering [28]. The optimal number of clusters was determined by varying a preference parameter that affects the a priori suitability of each data point to be a cluster center (**Fig 5A**, left panel). The optimal estimate of number of clusters (53 key features) falls in the knee of the curve, representing a balance between maximum compression and optimal cluster assignment accuracy (see Methods for details).

Additionally, we verified whether selecting only 53 key BO-NN features, as a result of clustering, still resulted in a high accuracy of predicting the neural responses (**Fig 5A**, right panel). Notably, most key features included electrodes from about half of all subjects and thus were not subject-specific.

Next, we evaluated the interpretability of all 53 key BO-NN features using previously mentioned tools (visualization, cortical projections, relation to known sound features, etc.).

**Visualization of key BO-NN features.**   First, we examined key BO-NN features that showed on average higher activation during either speech or music fragments (**Fig 5B**). We visualized the activity of some of these features (**Fig 5C**). Overall, multiple key features tracked audio changes visible on the spectrogram, such as silence versus sound moments and changes in intensity and difference between speech and music fragments. Statistical testing showed that some of the key BO-NN features were specific to speech (e.g. key features #1, #2 and #36), some were sensitive to both speech and music (e.g. key feature #42) and some showed an enhanced response to music (key features #48, #40 and #15), as assessed with a d-prime ($d'$) statistic for signal separability [29] (**Fig 5B,** see Methods for more details). At the same time, some features exhibited a complex pattern of activation across the sound types. For example, key feature #35 showed a sustained and slowly changing response to sound, often climbing throughout a music block and dropping throughout a speech block.

**Neural tuning and cortical maps per key BO-NN feature.**   Next, we examined the cortical weight maps of individual key BO-NN features (β-weights of the linear fit to the ECoG responses) and observed that multiple features contributed to the prediction of the time course of an individual electrode (**Fig 6A**).

The spatial distribution of the highest cortical weights (β-weights > 2 per each key feature) revealed that the perisylvian cortex and its adjacent regions were most associated with activity of the key BO-NN features (**Fig 6B**). We also visualized the cortical maps for some of the previously mentioned individual features with specific responses to speech and/or music (**Fig 6C**). The positive weights expressed a positive contribution of a key BO-NN feature to the prediction of the corresponding electrode, whereas the negative weights were associated with a reverse relationship.

Electrodes surrounding the Heschl's gyrus exhibited higher weights for the key feature that was active during both speech and music (key feature #42). Key features #1 and #2 showed distinct maps over the perisylvian cortex and both were most active during the speech fragments.

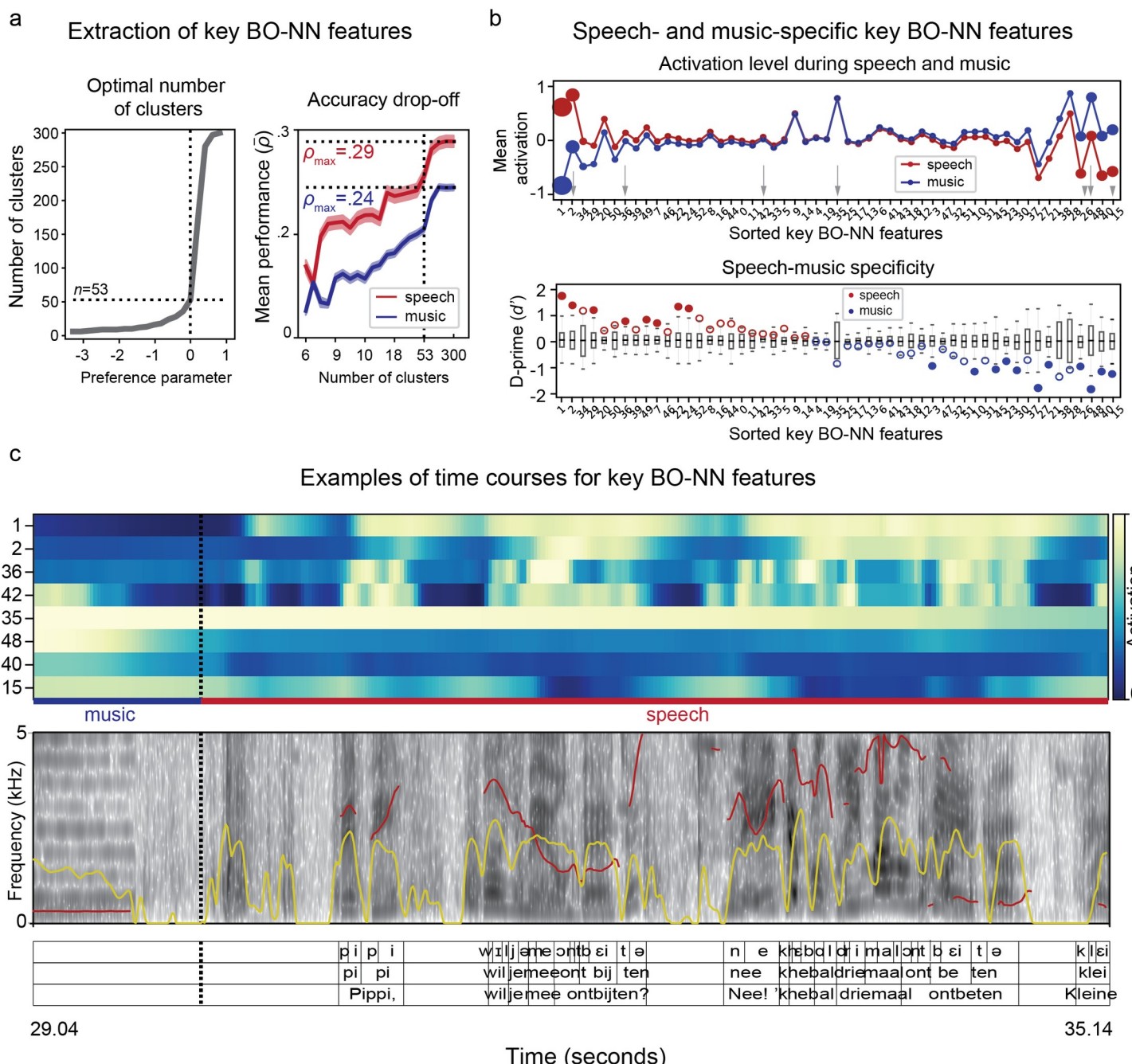

**Fig 5. Visualization and interpretation of the key BO-NN features. (a)** Optimal choice of the number of key features using AP clustering selected as the knee of the curve over the parameter of *preference* (left panel). The drop-off in the prediction accuracy is shown as a function of a number of clusters (right panel). The accuracy for predicting the neural responses to speech and music is shown separately. The prediction accuracy averaged over all significant electrodes is shown. Shaded area shows the standard error of the mean. **(b)** Top plot shows the key BO-NN features with the maximal average activation across speech or music fragments. Bottom plot shows music and speech specificity values per feature as assessed with the $d'$ statistics (signal separability index). Boxplots show surrogate distributions used for significance testing (obtained by permuting speech and music blocks and recalculating $d'$ values per feature 10000 times). The boxes show the 25th and 75th percentiles of the surrogate $d'$ values computed on permuted feature activation time courses, caps show 1th and 99th percentiles. A solid line in the middle shows the median. The actual $d'$ statistics (from non-permuted data) are shown as circle markers per feature. The markers are filled if the actual $d'$ statistics fall above the 99th (red markers, speech specificity) or below the 1st (blue markers, music specificity) percentile of the surrogate $d'$ distributions. **(c)** Example of a ~4-second fragment of activity for a number of key BO-NN features with the corresponding audio spectrogram and language annotations. The top three selected features (#1, #2 and #36) were most active during speech blocks (and exhibited the specificity to speech), whereas the bottom three selected features (#48, #40 and #15) were most active during music blocks (and exhibited the specificity to music). Feature activation values are the result of the tanh-transformation and are therefore in the range of [−1, 1]. Black dotted line shows the border between music and speech blocks. Yellow contour shows sound intensity, red contour shows pitch. Both were extracted automatically from Praat.

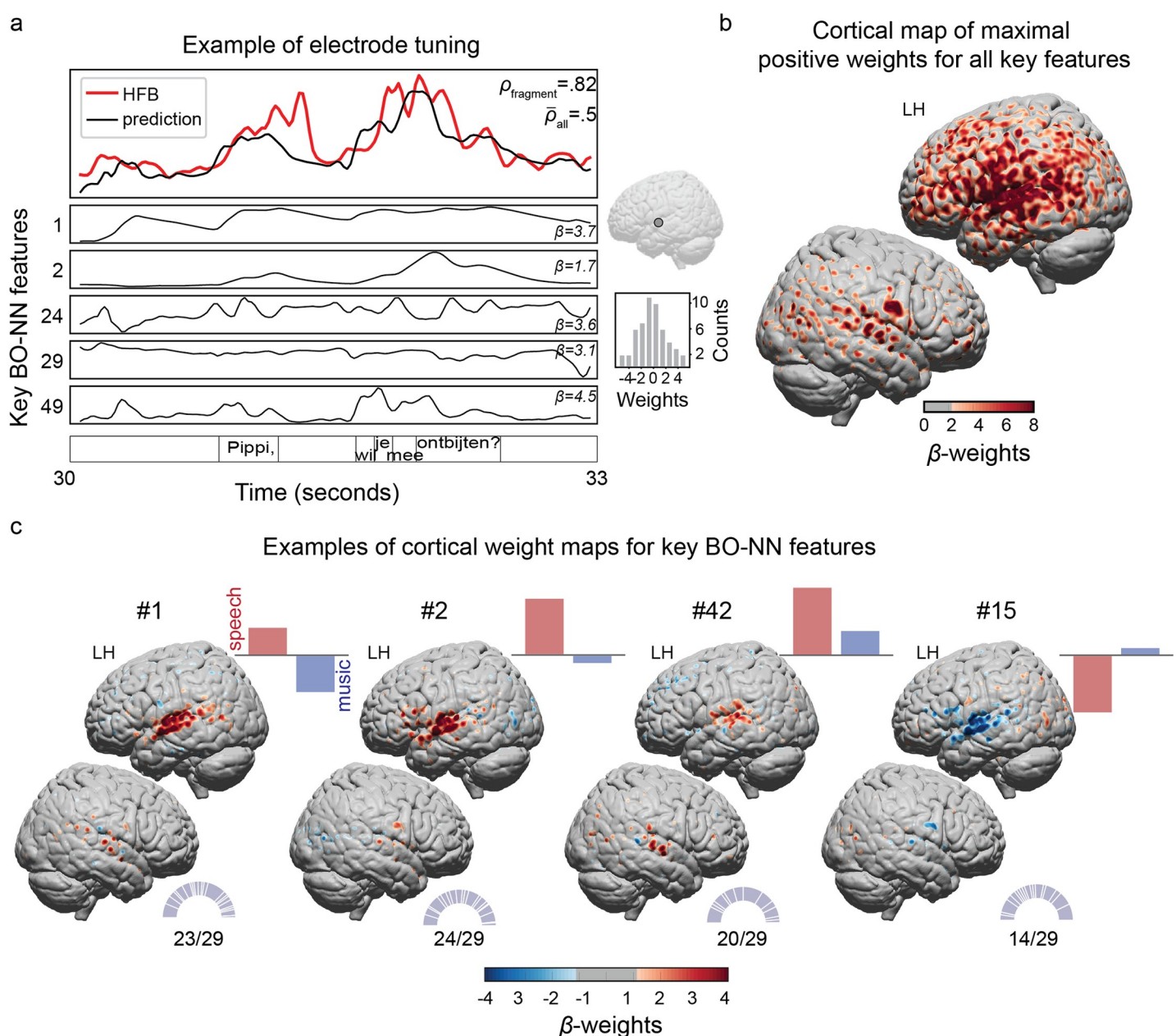

**Fig 6. Visualization and interpretation of the key BO-NN features. (a)** Example of an electrode tuning to various key BO-NN features. Top right panel shows the location of the selected electrode on the MNI cortical surface. Bottom right panel shows the distribution of $\beta$-weights over the 53 key BO-NN features for the selected electrode. **(b)** Cortical map of the maximal positive $\beta$-weights across all key BO-NN features highlighting that most electrodes associated with the time courses of the key BO-NN features were located in the perisylvian cortex and primarily on the left (note the left hemisphere bias in ECoG coverage). **(c)** Cortical weight maps over the electrodes for a number of the key BO-NN features. Per plot, bars in the top right corner show normalized mean activity of the feature during speech and music blocks, half-pie charts in the bottom right corner show the distribution of the electrodes contributing to the corresponding cluster over all 29 subjects.

Key feature #15 was more active during the music fragments compared to speech and showed a positive relationship with activity of the electrodes in the dorsal lateral occipitotemporal cortex (LOTC). This result likely reflects the interaction between the visual and auditory streams of Movie II (similar to Movie I). However, this interaction likely occurred at a higher level of conceptual representations given the cortical distribution for this key BO-NN feature and the

fact that there was no significant difference in the mean low-level luminance values between the music and speech fragments ($r^2 = .07$, nonsign.) and the lack of any difference in the rate of change (derivative) of the luminance values between the two sound types ($r^2 = .02$, nonsign.).

Notably, key feature #2 was associated with negative weights over dorsal LOTC. In fact, the cortical profiles of key feature #2 and key feature #15 seemed to oppose each other. The activation time courses of the two key features were largely anticorrelated during the speech fragments (Spearman's $\rho = -.82, p < .001$). During the music fragments the anticorrelation was reduced (Spearman's $\rho = -.41, p < .001$), suggesting that the relationship between the key features was modulated by the difference in speech and music.

**Acoustic patterns captured in key BO-NN features.** Next, we sought to interpret the key features through the acoustic sound properties they could have learned. First, we observed that the theory-driven STMFs were largely captured in the key BO-NN features, based on the linear regression models fitted to predict STMFs from BO-NN features (**Fig 7A**).

To obtain more detail we examined significant Spearman correlations of each key BO-NN feature with all STMFs, separately for speech and music (Movie II). In total, about a quarter of all key BO-NN features (14/52 in speech and 18/52 in music) showed large significant correlations ($\rho > .6$) with STMFs (**Fig 7B**). In the case of speech, the dimension with most variance in correlation profiles across the key BO-NN features was TMs (**Fig 7C**). There was little variance along the dimension of SMs. The correlation to STMFs along the frequency dimension varied within 250–3000 Hz (typical range of formants).

In the case of music, the correlation profiles varied most along the frequency dimension (**Fig 7C**). There was more variance in correlations to SM features compared to speech but less variance in correlations to TMs compared to speech. Most features were correlated with STMFs in a low frequency range of 100–700 Hz and fine SMs in range of $> 1$ cycle/octave. Correlations to TMs were mostly limited to $< 4$ Hz.

In general, multiple key BO-NN features captured complex multidimensional acoustic profiles. The combinations across the TM-SM-frequency dimensions contained essential information characteristic of each type of sound. In music, we observed the retaining of rich frequency-specific information, which changed slowly over time (slow TMs). In speech, we observed the retaining of rich TM information in a range of frequencies typically associated with formants.

## Distinct temporal profiles of key BO-NN features

It has previously been shown that the neural responses to sound are characterized by multiple temporal and latency profiles [4], therefor we investigated the temporal dynamics of the key BO-NN features as well. First, we looked at the autocorrelation profiles of the key BO-NN features and found that some of them showed slow-wave responses whereas others showed faster temporal changes (**Fig 8A**). Most key features exhibited fast temporal profiles with autocorrelation dropping within 50–100 ms. Only a small set of key BO-NN features exhibited slow temporal profiles with high autocorrelation over 200 ms. These features were associated with cortical weight maps outside STG with coverage of frontal or MTG electrodes.

Focusing on perceived speech processing, we used basic speech properties, such as intensity, pitch, formant information and a binary vector of speech being present or absent in the sound (speechON) to predict the key feature activations during speech at different time lags (see Methods for details). Most key BO-NN features showed tuning to intensity, pitch and speechON at different time lags (**Fig 8B**). Best predictions were typically obtained at 100–150 ms of sound onset.

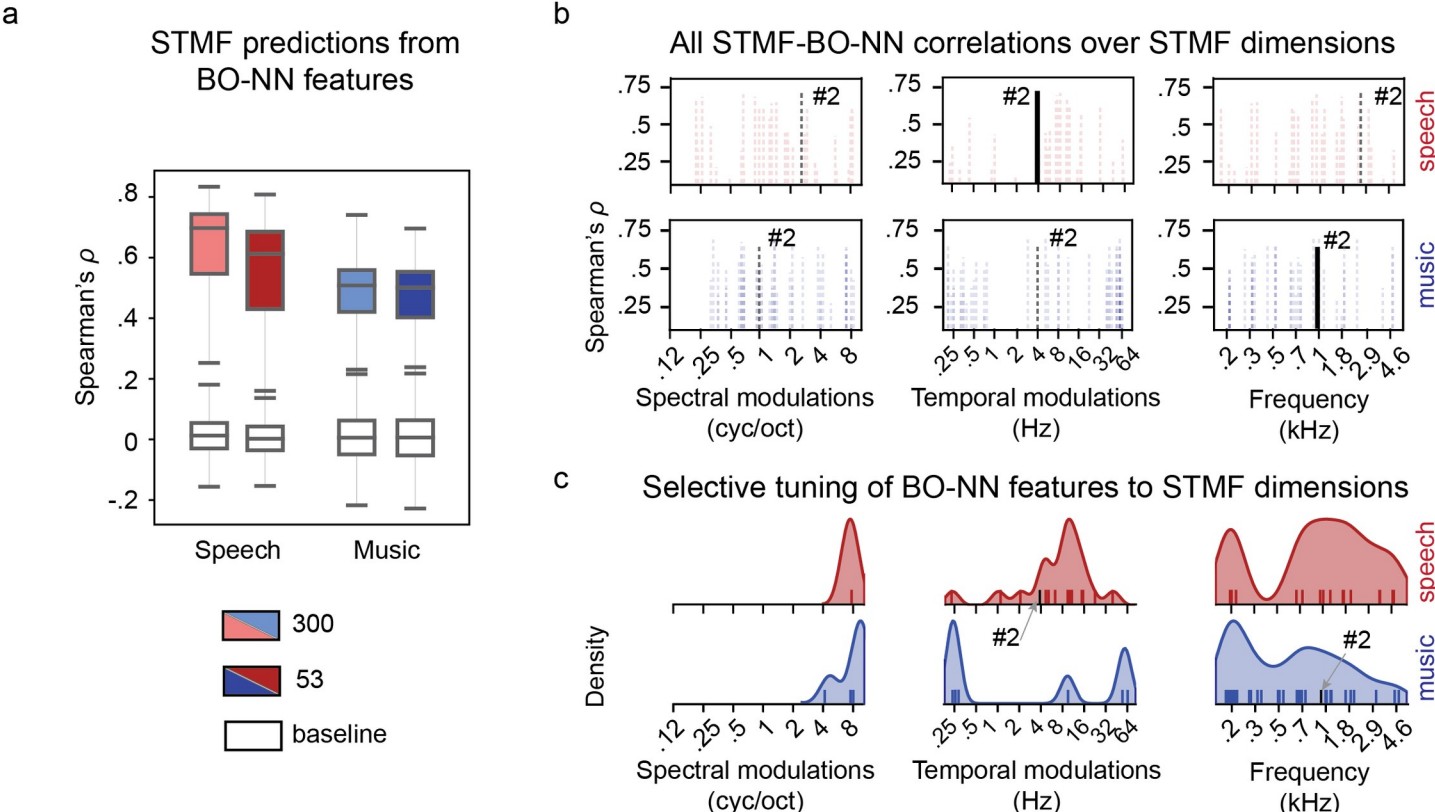

**Fig 7. Relation of the key BO-NN features to STMFs. (a)** Cross-validated prediction accuracy (over six folds) for modeling STMFs from the top BO-NN features, separately for speech and music, using either a full set of 300 top BO-NN features or only 53 key BO-NN features. The prediction accuracy was estimated as the Spearman correlation ($\rho$) between the predicted and observed STMF time courses in a held-out test set. Shown are Spearman's $\rho$ values significant at $p<.001$ (based on a surrogate distribution of shifted data, see Methods for details), averaged over six cross-validation folds. Colored boxes show the prediction accuracy for aligned BO-NN-STMF data during speech (in red) and music (in blue) fragments. White boxes show the distribution of the prediction accuracy for the surrogate shifted data (the alignment between the STMF and BO-NN time courses was shifted 1000 times). The boxes show the 25[th] and 75[th] percentiles of the prediction accuracy over all STMFs, the caps show 5[th] and 95[th] percentiles. A solid line in the middle shows the median. **(b)** Distribution of highest Spearman correlations with STMFs over all key BO-NN features, separately for speech and music fragments. Dotted lines show $\rho$ values per key BO-NN feature significant at $p<.001$ (based on a surrogate distribution of shifted data). Each key BO-NN feature with a significant $\rho$ value best captures a certain STMF that is a combination of values along three STMF dimensions: SMs, TMs and frequency. Therefore, per key BO-NN feature we mapped its maximal significant $\rho$ value in all three STMF dimensions. For example, key BO-NN feature #2 exhibits $\rho_{max}$ = .71 with a STMF that is a combination of SM = 2.8 cyc/oct, TM = 4 Hz and frequency = 2.7 kHz. This $\rho$ correlation is then shown at the corresponding values along each STMF dimension. **(c)** In addition, we investigated which of the three STMF dimensions each key BO-NN feature is selectively tuned to. For this, per key BO-NN feature we calculated the variance in BO-NN-STMF correlations along each STMF dimension and marked the dimension of the largest variance (see Methods for more details). Exhibiting preference for certain values along a specific STMF dimension was used as an estimate of dimension selectivity and tuning to a specific value along that dimension for each key BO-NN feature. These distributions of dimension selectivity for all key BO-NN features with significant correlations to STMFs are shown in the rug plots and the probability density plots above the correlation profiles per STMF dimension and sound type. Thus, the distribution plots show information that is different from the information shown on the correlation plots (dotted lines in **b**). In the example of the key BO-NN feature #2 the dimension selectivity is shown as a solid line in the correlation plot in **b**. It is also displayed in the rug plots and the probability density plots in **c**. The plots show that during speech the majority of the key BO-NN features were selective to the TM and frequency dimensions (as there was more variance in correlation for the different STMF values along these dimensions). Whereas the SM dimension, while showing high BO-NN-STMF correlations in **b**, was associated with less variance in the BO-NN-STMF correlations and therefore–lower preference in the key BO-NN features to its specific values.

Some features did stand out in this analysis. For example, key feature #41 showed a peak response to speech intensity -50–0 ms prior to the onset. Key feature #1 was most tuned to speechON at a lag of 200–350 ms, and key feature #2 –at a lag of 250–300 ms. Some features were associated with negative weights for speech being on and sound intensity in general, and thus were "deactivated" by speech (key features #15 and #26 associated with a weight map over the dorsal LOTC). Notably, the autocorrelation profiles and the preferred latency were decoupled for many key features (Spearman correlation over all 53 features between the time point

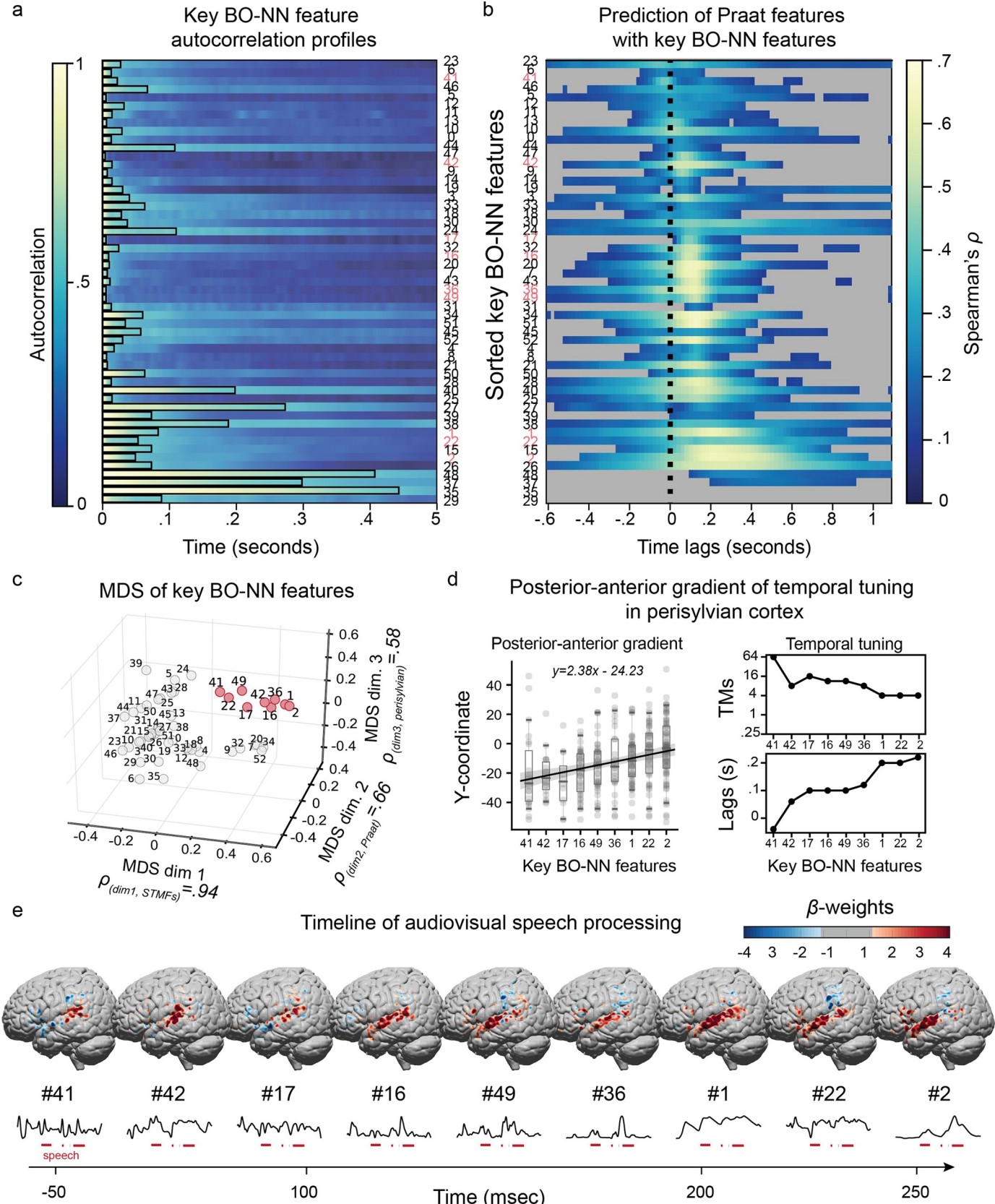

**Fig 8. Temporal profiles of the key BO-NN features. (a)** Autocorrelation profiles of 53 key BO-NN features. Per feature, a black line shows the time point where the autocorrelation drops below .5. **(b)** The accuracy of predicting the activity of the key BO-NN features from a set of speech features: speechON, sound intensity, pitch and frequency values for the first and second formants. A linear model was trained per each audio-ECoG shift in the range of -600 ms to 1 s. Spearman correlations between the predicted and observed activity of the key BO-NN features were calculated and averaged over the six cross-validation folds. The shifts associated with non-significant prediction accuracies are greyed-out. **(c)** The results of the MDS for the key BO-NN features. Various types of information were used for the MDS including the results of the previous analyses (correlation to STMFs and linear fit using Praat features), the magnitude of the associated cortical weights ($\beta$-weights) and the anatomical constraints (association with the cortical weights specifically from the perisylvian regions). Each circle represents one key BO-NN feature; and the labels correspond to the indices of the key features. The cluster of features with the highest values along all MDS dimensions are highlighted in red. Per MDS dimension we also show its maximal correlation with the specific type of information used in the analysis (e.g. correlation with STMFs). **(d)** Left plot shows the results of the ordinary least squares fit of the sorted key features (highlighted features in **c**) to the y-coordinate (anterior-posterior direction) of the center of mass for the ECoG electrodes associated with each feature (based on the $\beta$-cortical weights). The boxes show the 25th and 75th percentiles of the electrode y-coordinates per feature, caps show 5th and 95th percentiles. A solid line in the middle shows the median. The line across the boxplots represents the result of the ordinary least squares fit. The slope and intercept of the fit are also reported on the top of the plot. Right plot show the gradual decrease in the TM tuning for the features highlighted in **c** as well as the gradual increase in the temporal response profile (i.e. optimal shifts for the prediction of the key BO-NN features using Praat features shown in **a**). **(e)** A tentative timeline of the neural processing of the audiovisual speech in -50 to 250 ms around the sound onset. The plots contain cortical weight maps for the selected key features (highlighted in **c**) together with their example time course during a speech fragment. Only the electrodes from the perisylvian cortices are shown; the weights of the electrodes in the right hemisphere are mapped onto the analogous location in the left hemisphere.

when the autocorrelation drops below .5 and the latency of tuning to basic speech properties: $\rho$ = .2, $p$ = .17).

**Posterior-anterior gradient of temporal information in the perisylvian cortex during speech perception.**   Finally, we attempted to combine together the results we had obtained from all previous analyses aimed at interpreting the BO-NN features. We constructed a matrix that per key BO-NN feature contained the information about its correlation with the STMFs, its prediction accuracy from the linear fit using the Praat features and the magnitude of $\beta$-weights from the linear fit from top BO-NN features to the ECoG responses. We also added information regarding the cortical distribution of the electrodes associated with each key feature (through the $\beta$-weights of the linear regression fitted earlier), namely the percentage of the electrodes in the perisylvian cortices and the ratio of the electrodes in the perisylvian cortex to the electrodes in other brain areas. We focused on the perisylvian cortex because the prediction accuracy in these regions was best for the BO-NN model compared to the control models (**Fig 4**) and the majority of the electrodes that were associated with the BO-NN features were located there (**Fig 6B**).

Then, we used the multidimensional scaling (MDS) to visualize the space of the key BO-NN features according to the results of our previous analyses and the relationship to the activity in the perisylvian regions (**Fig 8C**). We observed that each MDS dimension was highly interpretable: MDS dimension 1 was most related to the correlation with STMFs during speech fragments ($\rho$ = .93, $p$ = 2×$10^{-24}$), MDS dimension 2 was most related to the results of the fit on Praat speech features ($\rho$ = .66, $p$ = 9×$10^{-8}$) and MDS dimension 3 was most related to the anatomical constraints and the relation to the brain activity in the perisylvian regions ($\rho$ = .58, $p$ = 4×$10^{-6}$). As a result of this projection, we also observed that a subset of key BO-NN features exhibited high values along all three MDS dimensions: #41, #42, #17, #16, #49, #36, #1, #22 and #2 (**Fig 8C**).

Previously we observed a trend for an increased latency of response for these features during speech perception: from -50–0 ms prior to speech onset (key feature #41) towards 250 ms after speech onset (key features #1, #22, #2, **Fig 8C**). Looking at the cortical maps of these features, we could also observe a trend for the electrode locations to move anteriorly along STG towards IFG. Therefore, next we tested the hypothesis about the posterior-anterior gradient of temporal information encoding along the perisylvian cortex [4,30,31]. We sorted the selected BO-NN features (highlighted in **Fig 8C**) according to their temporal response profiles (**Fig 8B**) and used them to fit the y-coordinate (posterior-anterior direction) for the center of mass of

the cortical weights within the perisylvian regions. This fit was significant: F(313,1) = 28.02, $p = 2\times10^{-7}$ (ordinary least squares fit, **Fig 8D**), supporting the idea of the posterior-anterior anatomical progression for the BO-NN features that were most correlated to acoustic STMF characteristics, were well fitted by sound intensity, pitch and speechON features, and were localized in the perisylvian cortical network. Importantly, this posterior-anterior gradient was associated with tuning to increasingly slow TMs (**Fig 8D**), providing data-driven evidence for the theory of the temporal gradients along the perisylvian regions involved in speech perception (**Fig 8E**).

## Discussion

In the present study, we succeeded in constructing a model of auditory perception in the human brain by using a bottom-up approach while abstaining from injecting specific theoretical assumptions about the sound features. We trained a model on one set of data and subjects, and then validated it in new data from different subjects. We found that a deep ANN, trained on sound input to directly predict the associated neural responses, can provide such a model. This model was best at predicting the cortical responses across lower-level and higher-level auditory processing cortices and it generalized well to unseen subjects and unseen audio data. It uncovered a number of features, interpretable in terms of their acoustic, cortical and response latency profiles. Certain features were activated specifically by speech and reflected encoding of hierarchical temporal structures across the perisylvian cortex. Altogether, we argue that a data-driven approach with a minimum of theoretical assumptions provides a powerful model that can be used to test existing theoretical assumptions and generate new insight about the complex mechanisms of auditory perception in the brain.

Similarly to the previous work [13,32], we observed that a hierarchical data-driven model, such as a deep ANN (BO-NN) can produce high accuracy of predicting the neural responses to complex sensory input. Moreover, we showed that a deep ANN with both convolutional and recurrent nodes, processing information over large audio fragments (3- to 6-second long) was best at fitting the associated cortical responses in our data. This result suggests that both short-term integration of sound (local convolutions) and analysis of long-distance temporal dependencies (recurrent processing) may play an important role in the cortical processing of auditory input. These findings extend the previous work on computational modeling of the neural auditory processing and resonate with the results of the ANN-based modeling of the brain responses during visual processing. Specifically, both convolutional [10,11,33] and recurrent [34,35] connections have been reported to play an important role in extracting the visual representations that closely match the associated neural responses. In addition, the improvement of the deep convolutional model over a shallow one is in line with the previous work [10,13,32] and offers further evidence of the hierarchical neural processing of the perceptual input [36,37].

Importantly, we were also able to show that the performance of our data-driven model generalized well to a different experimental setup (different movie-watching experiment) and a large, different, set of participants. On unseen data the prediction accuracy achieved by the BO-NN model exceeded the accuracy achieved with a theory-driven STMF-based model [4] or with an ANN model of identical complexity but not optimized on the neural data. Notably, the superior prediction accuracy achieved by the BO-NN model could not be reproduced simply by aggregating STMFs over various temporal delays. At the same time, we would argue that the ANN's inherent ability to learn the temporal function of the neural response directly from the data is in itself superior to the methods that require aggregation of features at multiple lags.

Importantly, the data-driven BO-NN model did not only lead to high accuracy of predicting the brain data, but it also provided complex data-driven features that led to the optimal predictions. In the present study we used feature visualization, clustering, correlation to known theory-driven features, and examination of cortical and temporal profiles, to gain insight into what information is captured in each data-driven feature. These features carry the potential to elucidate the nature of the relationship between perceived sound and the brain.

The power of the entirely data-driven approach was exemplified by the fact that the model extracted complex acoustic patterns that have been identified previously with a theory-driven model (SMs, TMs and frequency) [3,38]. Some of the extracted features were sound-specific (either speech or music), suggesting that they could distinguish between types of sound, reflecting a capacity for categorization. We have not yet observed a clear-cut distinction between low-level (acoustic properties) and higher-level (language or music types) features. The transformation from one to the other could be gradual and thus requires further examination of the BO-NN features, perhaps along the different layers of this deep ANN model.

Combining the information about the cortical distribution with the feature-specific acoustic and temporal latency profiles, we found an agreement with the theory of hierarchical processing of auditory input. In particular, the BO-NN model provides supporting evidence for hierarchical encoding of temporal information along the perisylvian cortex during speech perception [19,30,39,40], and for the notion of a posterior-anterior direction of temporal information propagation during naturalistic speech processing (**Fig 8E**) [4,30]. Moreover, our results agree with the concept of multi-scale temporal processing of complex stimulus events in the human brain [19,20]. Similar to the previous reports on speech perception, we observed that neuronal populations in posterior STG encode fast TMs (> 10 Hz), whereas sites along the sylvian fissure in the anterior direction up to the IFG encode increasingly slower TMs (< 4 Hz).

Some of the present findings also suggest that the posterior STG area, surrounding the Heschl's gyrus, may activate prior to, and at the moment of, speech onset. The effects of predictive coding [41] have previously been demonstrated in auditory perception, with similar spatiotemporal activity patterns for expected and perceived sounds [42,43]. Our results suggest the presence of a neuronal component in posterior STG that may be active prior to speech onset in a continuous speech stream within a complex audiovisual narrative. Our finding provides additional support that the observed phenomenon is not necessarily due to the prediction violation mechanism, typically observed in studies with stimulus mismatch/omission paradigms [43], but more likely indicates anticipatory effects inherent to continuous speech perception.

Additionally, we observed involvement of the dorsal LOTC regions in the sound, and particularly speech, processing network. This region responded to speech onset the latest (300–400 ms) and comprised a neuronal component that was deactivated by speech onset (key features #10, 15 and 26). These features did not correlate to acoustic sound properties and correlated negatively with the time courses across the perisylvian cortex. This anticorrelation decreased substantially in the absence of speech. One possibility is that the observed result was due to the interaction of the visual and auditory domains in the movie. For example, it is possible that the moments of speech aligned more with relatively static frames that likely contained people's faces, whereas music was more often present during scene changes, action scenes and frames without people in them. We believe that these distinctions are unlikely to represent low-level visual confounds, but rather reflect high-level interactions between the perception channels that are inherent in naturalistic stimuli, such as feature films. The apparent suppression of the visual motion information processing during speech can also be linked to the top-down processes as It has previously been shown that activation in this region can be modulated

by attentional demands in the context of audiovisual stimulation [44]. More work focusing on the audio-visual interactions as well as more comprehensive control for the low-level visual confounds will be needed to add to this discussion.

Not all data-driven BO-NN features were characterized by a relation to acoustic properties of sound. Several key features were informative for prediction of the signal in various cortical regions without a significant correlation to the STMFs. In addition, a number of features were characterized by slow-fluctuating time courses with a high autocorrelation rate and location in frontal brain regions. It remains to be further explored whether the extracted features can relate to other higher-level cognitive processes during movie watching, such as attention shifts, semantic information in the auditory stream and information of the movie plot present in both audio and visual modalities. It would also be interesting to explore how the attentional mechanisms affecting the cortical responses (for example, the observed difference between speech and music predictions in Movie II) could be integrated in the BO-NN model and learned as part of an explicit attention module. In addition, more work is needed to explore the multi-layer audio representations, and not only the layer closest to the neural predictions (top BO-NN layer).

Overall, a combination of the neural responses during a naturalistic stimulus experiment with a bottom-up modeling approach allowed us to build a generalizable, top-performance neural model of auditory perception with a set of interpretable key features that capture multi-level sound properties. The current pretrained model may be directly applicable to predict ECoG responses in new datasets using either controlled or naturalistic stimuli. The BO-NN model can also be further fine-tuned to fit the neural responses from another experimental setup. For example, given a previously demonstrated positive relationship between HFB and fMRI responses [45,46], it can be of interest to apply this framework in the context of fMRI data and explore tuning in the Heschl's gyrus, primary visual areas and the medial parts of the human brain which are inaccessible to ECoG grids.

## Conclusions

In the present study, we constructed a data-driven neural model of auditory perception by training a deep learning model on raw audio input to directly predict the associated brain responses, and validated it in new data from different subjects. Our brain-optimized model successfully fitted the brain responses along the perisylvian cortex and its adjacent regions. The extracted features captured complex spectrotemporal sound properties and exhibited distinct cortical distributions with specific latencies of response to sound onset. Several model features captured a temporal hierarchy of speech processing along the perisylvian cortex.

## Methods

### Ethics statement

The study was approved by the Medical Ethical Committee of the Utrecht University Medical Center in accordance with the Declaration of Helsinki (2013).

### Movie stimuli

A Dutch feature film "Minoes" (2001, www.bosbros.com) was used as stimulus for the first movie-watching experiment (Movie I). The film was 93 minutes long (78 minutes before credits) and told a story about a cat Minoes that one day transforms into a woman. In her human form, she meets a journalist Tibbe. Together, they solve several mysteries involving their town, and during their adventures eventually fall in love. The movie was made in Dutch and was easy to follow for both kids and adults. All patients reported that they enjoyed watching the movie.

In the second movie-watching experiment we used a 6.5 minute short movie, made of fragments from one of the Pippi Longstocking movies, namely "Pippi on the Run" (original title: "På rymmen med Pippi Långstrump", 1970) edited together to form a coherent plot (Movie II). The task was part of the standard battery of clinical tasks performed with the purpose of presurgical functional language mapping. The movie consisted of 13 interleaved blocks of speech and music, 30 seconds each (seven blocks of music, six blocks of speech). The movie was originally in Swedish but dubbed in Dutch.

## ECoG experiments

All participants were admitted for diagnostic procedures with medication-resistant epilepsy. They underwent subdural electrode implantation to determine the source of seizures and test the possibility of surgical removal of the corresponding brain tissue for about a week. Research could be conducted between clinical procedures. All patients gave written informed consent to participate in accompanying electrocorticography (ECoG) recordings and gave permission to use their data for scientific research.

Eight patients (age 22±7, six females) watched Movie I. One patient was excluded from the analyses because no electrodes covered language regions (no significant response to language or audio tasks). Another patient was excluded due to severe and frequent seizures. Thus, data from six patients was used. Three patients were implanted with left hemispheric grids. Most patients had left hemisphere as language dominant, based on fMRI or the Wada test (**Table 1**).

Thirty-three patients (age 26±13, 21 females) watched Movie II. Of those, four were excluded from the analyses because those patients also participated in the experiment with Movie I. Of the remaining 29 patients, 23 were implanted with left hemispheric grids. Most patients had left hemisphere as language dominant, based on fMRI, Wada or functional transcranial Doppler sonography test (**Table 1**).

All patients were implanted with clinical electrode grids (2.3 mm exposed diameter, inter-electrode distance 10 mm, between 48 and 128 contact points); one patient (experiment with Movie II) had a high-density grid (1.3 mm exposed diameter, inter-electrode distance 3 mm). Almost all patients had perisylvian grid coverage and most had electrodes in frontal and motor cortices. The total brain coverage of the patients who watched Movie II can be seen in **Fig 3**. Patient-specific information about the grid hemisphere, number of electrodes, and cortices covered is summarized in **Table 1**.

In the movie-watching experiment (Movie I or Movie II), each patient was asked to attend to the movie displayed on a computer screen (21 inches in diagonal). The stereo sound was delivered through speakers with the volume level adjusted for each patient. Due to the long duration of Movie I, the patients were given an option to pause the movie and quit the experiment at any time. In that case, the patient could continue watching the movie at a later time starting from the frame they had paused on.

During both experiments, ECoG data were acquired with a 128-channel recording system (Micromed) at a sampling rate of 512 Hz filtered at 0.15–134.4 Hz. Both movies were presented using Presentation software (version 18.0, Neurobehavioral Systems) and sound was synchronized with the ECoG recordings. Additional data, such as audio/video recordings of the room and PictureInPicture were used to ensure the synchronization.

## ECoG data processing

All electrodes with noisy or flat signal (visual inspection) were excluded from further analyses. After applying a notch filter for line noise (50 and 100 Hz), common average rereferencing was applied to all clinical grids per patient (and separately for the one high-density grid). Data

**Table 1. Electrode grid information for all participants (both Movie I and II).** Shown is information about the number of electrodes, grid hemisphere, covered cortices, handedness, and language-dominant hemisphere per patient. L, Left; R, right; F, frontal cortex; M, motor cortex; T, temporal cortex; P, parietal cortex; O, occipital cortex; fMRI, functional magnetic resonance imaging; fTCD, functional transcranial Doppler sonography.

| Patient | N of electrodes | Grid hemisphere | Cortices covered | Handedness | Language dominance | Movie experiment |
|---|---|---|---|---|---|---|
| 1 | 88 | R | M, T, P, O | R | L (fMRI) | I |
| 2 | 64 | R | F, M, T, P | R | L (fMRI) | I |
| 3 | 64 | R | T, P, O | R | L (Wada) | I |
| 4 | 64 | L | F, M, T, P | R | L (fMRI) | I |
| 5 | 64 | L | M, T, P, O | L | L (fMRI) | I |
| 6 | 120 | L | T, P, O | R | L (fMRI) | I |
| 7 | 96 | L | F, T, P | R | L (Wada) | II |
| 8 | 112 | L | F, M, T, P, O | R | L (fMRI) | II |
| 9 | 96 | L | F, M, T, P, O | R | L (Wada) | II |
| 10 | 104 | L | F, M, T, P | R | L (Wada) | II |
| 11 | 48 | L | F, M, T, P | L | L (fMRI) | II |
| 12 | 120 | L | F, M, T | R | L (Wada) | II |
| 13 | 112 | L | F, M, T, P | R | L (fMRI) | II |
| 14 | 64 | L | F, M, T | R | L (fMRI) | II |
| 15 | 112 | L | M, T, P, O | R | L (fMRI) | II |
| 16 | 64 | L | M, T, P | R | L (Wada) | II |
| 17 | 96 | R | M, T, P, O | L | R (Wada) | II |
| 18 | 88 | L | T, P, O | R | L (fMRI) | II |
| 19 | 112 | L | F, T, P | R | R (Wada) | II |
| 20 | 120 | R | F, M, T | R | L (Wada) | II |
| 21 | 96 | R | F, T, P, O | L | L (Wada) | II |
| 22 | 120 | L | F, T, P, O | not available | L (Wada) | II |
| 23 | 88 | L | F, M | R | L (fMRI) | II |
| 24 | 96 | L | F, M, T, P, O | L | L (Wada) | II |
| 25 | 64 | R | T, P, O | R | not assessed | II |
| 26 | 88 | L | F, M, P | R | L (fMRI) | II |
| 27 | 80 | L | F, M, T, P | R | L (Wada) | II |
| 28 | 128 | L | F, M, T, P, O | R | L (fMRI) | II |
| 29 | 80 | R | F, M, T, P, O | R | L (fMRI) | II |
| 30 | 64 | L | F, M, T | R | L (fMRI) | II |
| 31 | 64 | L | F, M, T | R | L (fMRI) | II |
| 32 | 48 | R | F, M, T | L | R (fMRI) | II |
| 33 | 112 | L | F, M, T | R | R (fMRI) | II |
| 34 | 64 | L | F, M, T | R | L (fTCD) | II |
| 35 | 64 | L | F, M, T, P | R | L (fMRI) | II |

were transformed to the frequency domain using Gabor wavelet decomposition over 1–120 Hz in 1 Hz bins with decreasing window length (4 wavelength full-width at half maximum). Finally, high frequency band (HFB) amplitude was obtained by averaging amplitudes for the 60–120 Hz bins and the resulting time series per electrode were down-sampled to 50 Hz. Electrode locations were coregistered to the anatomical MRI in native space using computer tomography scans [47,48] and FreeSurfer (http://surfer.nmr.mgh.harvard.edu/). The Desikan–Killiany atlas [49] was used for anatomical labeling of electrodes (closest cortical structure in the radius of 5 mm). All electrode positions were projected to Montreal Neurological

Institute space using SPM8 (Welcome Trust Centre for Neuroimaging, University College London).

## Audio processing

**Description of audio data in Movie I.**    We used a manual annotation to make a summary of the different sound fragments in the soundtrack of Movie I. From the film production company we obtained the film subtitles and film script. These were used to produce a preliminary text-to-audio alignment in Praat [50]. The alignment was created automatically by converting subtitle text into Praat annotations based on the subtitle time stamps. The subtitle text was then compared against the script and corrected accordingly. Next, a number of undergraduate students were employed to correct the automatic text-to-audio alignment. Each student corrected the time markers of the subtitle text and created a tier with markers for onsets and offsets of individual words. The students received detailed instructions regarding the waveform and spectrum properties of sound that could aid in determining the onsets and offsets of individual words. A trained linguist further verified their manual annotation. The students also marked fragments containing different types of sound. As a result, we obtained a sound annotation file with multiple tiers: subtitle text (1), individual word boundaries (2), music (3), noise (4), animal cries (5), environmental sounds (6) and other sounds (7). "Environmental sounds" were distinguished from the "animal cries" as the movie plot involved many animals (specifically cats), which was relevant for another project. "Environmental sounds" therefore contained other sounds, such as rain, thunder, body sounds (clapping, footsteps), etc. The "noise" category included technical noises (car sounds, phone ringing, typing sounds), objects falling and other indistinguishable noises. "Other sounds" included sound fragments that were difficult to categorize, such as synthetic sound effects and sound beeps, which only constituted 3% of the soundtrack. The annotations in tier 2 (individual word boundaries) were used as speech labels. It is therefore important to note that during speech fragments all pauses between chunks of continuous speech (time gaps between word labels) were not considered "speech", but "noise", "sound", "ambient", etc. depending on the nature of the sound. Thus, there is little pause data in the fragments labeled as "speech". The overlap between the different types of sound was easily calculated from the overlapping annotations across the tiers. Any unlabeled fragments were considered to be 'ambient sounds'.

**Preprocessing of audio for BO-NN.**    For the "spectrogram stream" of the BO-NN model, the spectrogram was extracted from the raw audio at 16 kHz with 8 ms frames for 128 logarithmically spaced frequency bins in the range of 180–7000 Hz using the NSL toolbox [2] (see also the details of the STMF extraction below). For the "time-domain stream", the time domain data were downsampled to 8 kHz.

**Acoustic similarity of sound in Movie I and Movie II.**    To estimate how much we should expect the performance of the BO-NN model to generalize from the training data of Movie I to unseen audio material of Movie II we analyzed the similarity of acoustic features of Movie I and Movie II. First, we extracted spectrotemporal modulation features (STMFs) of audio in both datasets (Movie I and Movie II) using the NSL toolbox [2] (see the details below). The STMFs are the outcome of a 2D Gabor filtering of the sound spectrogram obtained by varying orientations and sizes of Gabor filters. We used this representation here and later in the paper as it is known to represent salient acoustic properties of sound [51] and has been shown to be captured in the neural responses [3,4,52]. We downsampled the STMFs of sound to 50 Hz and then selected a sound sample (one sound data point at 50 Hz) from Movie II at random. We computed the cosine similarity of its STMFs with STMFs for sounds of the same type (music or speech) in Movie I, noting down the value of the maximal cosine similarity. The procedure

was repeated 1000 times for music sound samples (only compared with music in Movie I) and then run separately (also 1000 times) for speech sound samples (only compared with speech in Movie I). As we did not have enough samples of "noisy", "environmental", "ambient" and "animal" sounds in Movie II the analysis was only limited to "speech" and "music". We assessed the difference in similarity across datasets for speech and music sounds using a Mann-Whitney signed-rank test on 1000 maximal similarity data points and found no significant difference in similarity across datasets between speech and music.

## Architecture and training of the BO-NN model

We employed various ANN architectures to train a model end-to-end by taking raw audio (sampling rate of 16 or 8 kHz) as input and predicting the HFB neural activity as output. HFB time courses were concatenated across all electrodes of six patients (396 electrodes in total). The input (audio) and output (HFB ECoG) data were first preprocessed using a standard scaler from *scikit-learn* [53] that z-scored the data per channel (per frequency bin of the spectrogram and per ECoG electrode in brain data).

To investigate how different architectural choices affected the model performance we trained several ANN models and compared their performance using non-parametric Wilcoxon signed-rank tests. Whenever possible, the ANNs were matched in terms of the number of parameters (on average approximately 3318000 parameters). In ANNs containing a convolutional component (CNN and RCNN), each convolutional layer was followed by a batch normalization layer, a rectified linear activation function and local pooling. Dropout [54] of 10% was used after each pooling layer. In ANNs containing a recurrent component (RNN and RCNN) we used Zoneout long-short-term-memory (LSTM) cells [55] for improved regularization. All networks were trained by minimizing the mean squared error using the Adam optimization algorithm ($\alpha = 5 \times 10^{-4}, \beta_1 = .9, \beta_2 = .999$). The models were constructed and trained using a deep learning framework Chainer [56], version 6.1.

Several ANN comparisons were performed. First, we compared ANNs with different computational mechanisms within the main nodes: a convolutional neural network (CNN), a recurrent neural network (RNN) and a recurrent convolutional neural network (RCNN). The RCNN first performed convolutions on the input data and then passed the output of the convolutional layer to the recurrent layer, which then projected to the output layer. All these ANNs were deep (each contained five layers) and were trained on the time-domain audio input. Second, we compared the ANNs with the respect to the type of input data: an ANN trained on the time-domain audio input, an ANN trained on the audio spectrogram and an ANN trained on a combination of the two. The latter ANN is also called a 'dual-stream' ANN as it processed both inputs separately at first and concatenated them later on (**Fig 1A**). Since the RCNN was the best model in the first comparison, we used the RCNN architecture to compare ANNs trained on different types of the audio input. Since there was no significant difference between the RCNN trained on the time-domain, the spectrogram or a combination of inputs, we chose the latter configuration for further comparisons. The reason for that was that we aimed to maximize the generalization capacity of the ANN and considered that both types of input may carry information that could aid generalization. Third, we compared a more shallow version of the RCNN (containing only one layer per input stream in the convolutional block) and a deeper version of the RCNN (containing four time-domain and five spectrogram-processing layers in the convolutional block). Fourth, we compared RCNNs that received one-, three- or six-second audio fragments as input. All three RCNNs were deep and took both time-domain and spectrogram data as input.

The best performing model had a deep RCNN architecture (combining convolutional and recurrent layers) processing six-second audio fragments at once, fed in as both time-domain and spectrogram representations. Similar architectures have been utilized before when training task-optimized ANNs [13]. The dual-stream model passed the 1D raw audio waveform (six-second input fragment at 8 kHz) through five 1D convolutional layers and the 2D audio spectrogram (six-second input fragment at 50 Hz)–through four 2D convolutional layers. Then it concatenated the output and passed it to the recurrent layer with Zoneout LSTM cells. The details of the BO-NN parameters (such as kernel size, pooling size and stride, padding, etc.) are summarized in **Fig 1A**. The comprehensive list is shown in **Table 2**.

## BO-NN model performance with Movie I (six participants)

Model performance was quantified as the Spearman correlation between predicted and observed HFB responses in the test set of Movie I (10% of all data):

$$\rho = \frac{\text{cov}(rx, ry)}{\sigma_{rx}\sigma_{ry}} \tag{1}$$

where $rx$ and $ry$ are rank-transformations of $x$ and $y$, respectively, and $\sigma_{rx}$ and $\sigma_{ry}$ are the standard deviations of the rank variables.

A ten-fold cross-validation was used for an unbiased estimation of the model performance. Per each test fold the training continued for 30 epochs (i.e. full runs through all training data). The model estimated at the epoch with the best test performance across all test folds was used later on for testing of performance generalization to a dataset of Movie II.

The significance of the prediction accuracy was assessed parametrically by transforming Spearman's $\rho$ scores to $t$-values (Eq 2) and determining the $p$-values based on the probability density of the Student's distribution. The reported $\rho$ scores were significant at $p < 1 \times 10^{-20}$, Bonferroni corrected for the number of electrodes and test folds.

$$t = \rho\sqrt{\frac{n-2}{1-\rho^2}} \tag{2}$$

To assess whether there was any difference in the model performance depending on the type of the predicted sound, we extracted and separated speech, music, noise and ambient sound fragments from each test fold based on the manual annotation of the soundtrack. The fragments of one type of sound were concatenated within each fold. The fragment length across different types of sound was balanced by subselecting fragments of 30 s per sound type and per fold. The prediction accuracy was calculated separately for each fold and averaged across the folds. Significance testing was performed in the same way as described above.

## BO-NN model performance with unseen data (29 new participants, Movie II)

The soundtrack of Movie II was passed through the pretrained BO-NN model to obtain the feature activations for a new dataset. The model weights were frozen and the model was not fine-tuned to optimize the predictions on the dataset of Movie II. Then, a ridge linear regression (Eq 3) was trained on each layer $l$ of the BO-NN model to predict the HFB responses (matrix $\mathbf{Y}$ of size $N_{timepoints} \times N_{electrodes}$) to the sound features $\mathbf{X}_l$ of Movie II:

$$\mathbf{Y}_l = \mathbf{B}_l^{\text{T}}\mathbf{X}_l + \boldsymbol{\varepsilon}_l \tag{3}$$

**Table 2. Parameters of the BO-NN model.** Conv, Convolutional layer; BN, Batch normalization layer; ReLU, Rectified linear units; LSTM, long-short-term-memory units. The table is divided into blocks corresponding to two streams of processing: CNN1 on time-domain input, CNN2 on time-frequency input (sound spectrogram), concatenation layer, RNN layer (LSTM) and the output layer that contains predictions for the HFB ECoG responses across electrodes (396 electrodes).

| NN block | Layer | Number of filters | Size of filter | Stride | Padding | Size of output |
|---|---|---|---|---|---|---|
| CNN 1 time: time × 1 | Padding 1_1 | | | | 750×750 along time | 1×49500×1 |
| | Conv 1_1 | 128 | 19×1 | 1×1 | 9×0 | 128×49500×1 |
| | BN 1_1 | 128 | | | | |
| | ReLU 1_1 | | | | | |
| | Pooling 1_1 | | 15×1 | 15×1 | 0×0 | 128×3300×1 |
| | Dropout 1_1 | | | | | |
| | Conv 1_2 | 128 | 15×1 | 1×1 | 7×0 | 128×3300×1 |
| | BN 1_2 | 128 | | | | |
| | ReLU 1_2 | | | | | |
| | Pooling 1_2 | | 11×1 | 11×1 | 0×0 | 128×300×1 |
| | Dropout 1_2 | | | | | |
| | Conv 1_3 | 256 | 9×1 | 1×1 | 4×0 | 256×300×1 |
| | BN 1_3 | 256 | | | | |
| | ReLU 1_3 | | | | | |
| | Conv 1_4 | 512 | 5×1 | 1×1 | 2×0 | 512×300×1 |
| | BN 1_4 | 512 | | | | |
| | ReLU 1_4 | | | | | |
| | Conv 1_5 | 300 | 1×1 | 1×1 | 0×0 | 300×300×1 |
| CNN 2 spectra: time × frequency | Conv 2_1 | 64 | 5×7 | 1×1 | 2×3 | 64×300×128 |
| | BN 2_1 | 64 | | | | |
| | ReLU 2_1 | | | | | |
| | Pooling 2_1 | | 1×5 | 1×5 | 0×0 | 64×300×26 |
| | Dropout 2_1 | | | | | |
| | Conv 2_2 | 128 | 5×7 | 1×1 | 2×3 | 128×300×26 |
| | BN 2_2 | 128 | | | | |
| | ReLU 2_2 | | | | | |
| | Pooling 2_2 | | 1×5 | 1×5 | 0×0 | 128×300×6 |
| | Dropout 2_2 | | | | | |
| | Conv 2_3 | 256 | 3×7 | 1×1 | 1×3 | 256×300×6 |
| | BN 2_3 | 256 | | | | |
| | ReLU 2_3 | | | | | |
| | Pooling 2_3 | | 1×7 | 1×7 | 0×0 | 256×300×1 |
| | Dropout 2_3 | | | | | |
| | Conv 2_4 | 100 | 1×1 | 1×1 | 0×0 | 100×300×1 |
| | Concatenation layer | 400 | | | | 400×300×1 |
| | Stateful Zoneout LSTM | 300 | | | | 400×300×1 |
| | Output layer | 396 | | | | 396×300×1 |

where $\varepsilon_l \sim \mathcal{N}(0, \sigma^2)$. Both the sound features extracted at each layer of the BO-NN model and the HFB responses to Movie II were z-scored prior to the ridge linear regression fit.

The optimal value of the regularization parameter $\lambda$ (Eq 4) was determined through a nested cross-validation ($k_{nested}$ = 10). All linear models were fitted using a machine learning Python package scikit-learn [53].

$$\hat{\mathbf{B}}_l = \mathrm{argmin}_{\mathbf{B} \in \mathbb{R}} \|\mathbf{Y}_l - \mathbf{B}_l^{\mathrm{T}} \mathbf{X}_l\|_2^2 + \lambda \|\mathbf{B}_l^{\mathrm{T}}\|_2^2 \qquad (4)$$

Because the dataset of Movie II comprised interleaved blocks of speech and music, we trained separate ridge regression models to predict HFB responses to speech and music separately. As before, the model performance was measured as the Spearman correlation between the predicted and the observed HFB responses in the test set. The model performance was cross-validated across six 30-second test folds. Thus, each sound block of speech or music was used as a test fold to minimize the effect of variance across the blocks.

The significance of the linear model performance was assessed using an exact test. For each model (each layer of the BO-NN model and separately for speech and music) we disrupted the alignment between the sound and the HFB responses and trained a ridge regression model using the misaligned time courses. Specifically, first we permuted the sound blocks in the input data (soundtrack of Movie II contained 13 interleaved blocks of speech and music) and additionally shifted the time course of the permuted input data to further disrupt the alignment with the HFB responses. The minimal shift was set to be 20 seconds to account for the autocorrelation in the data. Data shifting and retraining of the ridge regression model was repeated 1000 times per model and yielded a baseline distribution of accuracy scores (n = 1000). The significance threshold for the performance of the model with aligned time courses was set at the 99.999[th] percentile of the baseline shifted distribution, which corresponds to a *p*-value of .001.

The performance across layers of the BO-NN model was compared using one-sided Wilcoxon signed tests. The layer comprising LSTM cell states showed the highest prediction accuracy and was used in all further analyses. It is referred to as the top layer of the BO-NN model. The hidden states of the LSTM layer showed similar prediction accuracy.

## Comparison of the data-driven BO-NN features with the theory-driven feature set

As a state-of-the-art theory-driven feature set we used STMFs [2], which were previously used to model the HFB responses in a subset of subjects who watched Movie II [4]. The features of the modulation set are constructed based on a biological model for cortical processing of the time-frequency representation of sound. At the first stage, the auditory spectrogram of sound is calculated by modeling the auditory responses of the cochlea and the midbrain. This process involves a pass of the audio signal through a bank of frequency filters, followed by the transformation into the auditory nerve pattern through a non-linear compression and a membrane-leakage low-pass filter, followed by a pass through a lateral inhibitory network of the cochlea nucleus and signal integration with Gaussian windows mimicking the auditory processing in the midbrain.

At the second stage, the STMFs are computed from the previously extracted auditory spectrogram, which reflects the spectrotemporal analysis postulated to take place in the primary auditory cortex. The auditory spectrogram is decomposed using a bank of 2D filters of predefined sizes and orientations, with the two dimensions corresponding to temporal modulations (TMs) and spectral modulations (SMs), respectively. The resulting complex auditory representation captures the phase and magnitude of the modulation content within a defined range of frequencies and time points. For details of the computational mechanisms and the associated mathematical equations we refer the readers to the original paper on STMFs [2]. In the present work we used the STMF implementation provided in the NSL toolbox [2] and only changed the parameters configuring the STMF filters.

Based on the previous work [4,25,32], here we used the following ranges for extracting the STMFs: a range of 0.125–8 cyc/oct for SMs and a range of 0.25–64 for TMs. To ensure a fair comparison with top BO-NN features, we selected the best performing set of STMFs by varying

the size and number of the filters along both temporal and spectral dimensions as well as subsampling or averaging along the frequency axis. In total, we looked at eight various subsets comprising 63, 119, 221, 252, 936, 952, 1768 or 3536 features. The subsets also included features from reduced ranges of SMs (only > 1 cyc/oct) and TMs (only ≤ 4 Hz) that were subsampled or averaged over multiple frequency bins (1, 4, 8 or 16). The feature set selected for further comparisons contained 13 SMs in the range of 0.125–8 cyc/oct, 17 TMs in the range of 0.25–64 and eight frequency bins (resulting from averaging instead of subsampling), in the range of the original spectrogram: 180–7000 Hz. In total, this resulted in 1768 STMF features. This feature set achieved one of the highest accuracy scores in predicting the HFB neural responses.

To compare the representational geometries across different models and identify the feature set with the representational geometry most similar to the neural responses in STG we employed the representational similarity analysis (RSA) [26,27]. Within the RSA framework, internal representations of various candidate feature sets (STMFs, Rand-NN and top BO-NN) are compared to the target dataset (neural responses in STG) to assess which representation is least dissimilar to the target dataset. Here, we used Spearman correlation ($\rho$) across $N$ time points ($N \times N$ matrices, vectorized) as the internal representation of each dataset ($N \times F$, where $F$ is the feature dimension). This was calculated separately for the speech and music fragments of Movie II. The dissimilarity between the target and the candidate datasets was calculated as $1 - \rho$ of their internal representations.

The RSA was performed separately for the speech and music fragments to account for the possible difference in representation of different types of sound. All sound-specific fragments were concatenated in time and split into 6-second chunks (each containing 300 time points at 50 Hz), resulting in 30 speech and 30 music chunks. Then, per chunk, the first order dissimilarity matrices were constructed by computing $1 - \rho$ (Spearman correlation) across the 6 seconds: the correlation was computed between each feature set (candidate datasets) and the neural responses in STG electrodes (target dataset) across the time points of the chunk. All STG electrodes (199 in total) were used as features of the STG representation. The procedure resulted in 30 matrices (one per each 6-second chunk) of size $300 \times 300$ per dataset, each containing dataset-specific pairwise dissimilarities between time points within the chunk. Then, per chunk, the second order dissimilarity matrix was constructed by vectorizing values above the main diagonal (44850 values) from first order dissimilarity matrices (30 matrices per dataset, target and candidates) and computing $1 - \rho$ between the datasets. The resulting dissimilarity values ($1 - \rho$) were compared between the target dataset and each of the candidate datasets in all 30 sound-specific chunks for both speech and music. Essentially, in this approach each time point corresponds to a 'trial' and each 6-second chunk corresponds to a 'subject' in a standard RSA setting with the difference that the 'trials' (time points) are not the same across the 'subjects' (6-second chunks). In this setup, the brain data from real subjects with STG electrodes was aggregated together (199 electrodes in total) and treated essentially as coming from the same subject and thus making one (target) dataset. The closest match to the target dataset was determined via a non-parametric two-sided Wilcoxon signed test using a dissimilarity value per each chunk as a single data point.

Next, to compare the model performance across the feature sets in STG and higher-level cortical areas during speech we trained a ridge linear regression model to fit each electrode in STG, IFG, postcentral gyrus and MTG. As input to the linear regression we used 1768 STMFs, 300 Rand-NN or 300 top BO-NN features. The linear model training and evaluation were performed in the same way as described previously, by employing a nested cross-validation to estimate the regularization parameter and computing the prediction accuracy by cross-validating per-block performance. The cross-validated performance was compared between the three models in a non-parametric two-sided Wilcoxon signed test per ROI using each electrode's

performance as a single data point. Only electrodes with significant cross-validation performance in any of the three models (as determined by non-parametric testing based on shifted data, n = 1000, see above) were included in the analysis.

In addition, we trained another ridge linear regression model using STMFs at multiple temporal delays with respect to the audio onset to compare it against the ridge linear model using BO-NN features that did not incorporate any temporal delays. At each time point within 500 ms from the audio onset a vector of 1768 STMFs was added to the model following the setup from the previous studies [4,25]. At 50 Hz this resulted in 44200 features to predict the HFB responses per one time point. The linear models using both feature sets (STMFs at various temporal delays and BO-NN with no temporal delays) included a regularization parameter to improve the fit and generalization. The model training and evaluation were performed in the same way as described previously, by employing a nested cross-validation to estimate the regularization parameter and computing the prediction accuracy by cross-validating per-block performance. The cross-validated performance was compared between the two models using two-sided Wilcoxon signed test per ROI as described previously.

## Extraction of key BO-NN features

To minimize the number of features to work with, we identified the key features of top BO-NN using another data-driven approach, namely, affinity propagation (AP) clustering [28]. This approach offers a benefit of identifying key data points (cluster centers) that are most representative of the clusters they belong to. This means that there is no additional transformation from the original features to a new feature space. This is convenient because we can directly visualize the cortical weight map associated with each cluster center (key feature), as it is simply the weight map of the feature that already exists in the dataset. The optimal number of clusters was identified in a data-driven way by varying the parameter of *preference* that represents the initial suitability of each data point to be a cluster center. By default, AP clustering preference value is set to the median of the similarity matrix (or *affinity* matrix). Here, instead of using the default we varied the value of *preference* until the estimate of the resulting number of clusters fell in the knee of the curve, representing the optimal balance between maximum compression and the optimal cluster assignment accuracy.

AP clustering was performed on the electrode $\beta$-weight profiles, which were learned by the ridge linear regression that predicted HFB responses from the 300 top BO-NN features. The $\beta$-weights were averaged over the six cross-validation folds. We took the $\beta$-weights from the linear model that was tested on the speech fragments (with six-fold cross-validation over six speech blocks). Even though another regression was trained and subsequently tested only on the music fragments, during training both regression models were exposed to roughly the same amount and type of input data (the remaining 11 30-second blocks), and the $\beta$-weights did not differ significantly between them. AP clustering was applied to the $\beta$-weight profiles using the Spearman correlation matrix across the $\beta$-weights as the *affinity* matrix.

Additionally, we estimated the accuracy drop-off depending on the number of clusters separately in speech and music by using only the time courses of the cluster centers (that differed depending on the *preference* parameter) as predictors of the HFB responses. This was done to check that the reduction of the key BO-NN features did not dramatically affect the accuracy of predicting the neural responses in Movie II.

## Selection of speech-specific and music-specific features

Features with peak activation during speech and music fragments were identified by comparing their mean normalized activation levels across the fragments. In the case of speech, we

used manual annotations for word onsets and offsets rather than the entire speech block because of the presence of a large number of pauses in speech, whereas music was constantly present throughout the music blocks.

In order to test feature specificity to speech or music, we calculate the d-prime ($d'$) statistic [29] per feature. The statistic assesses the degree of separability of two signal distributions and in our case, the separability of speech and music activations per feature:

$$d' = \frac{\mu_s - \mu_m}{\sqrt{\frac{1}{2}\left(\sigma_s^2 + \sigma_m^2\right)}} \tag{5}$$

where $\mu_s$ and $\sigma_s$ are respectively the mean and the standard deviation of the feature's activation during speech fragments, and $\mu_m$ and $\sigma_m$ are respectively the mean and the standard deviation of the feature's activation during music fragments.

We then performed the significance testing by constructing surrogate distributions of $d'$ values per feature computed on permuted activation time courses. For that, speech and music blocks were permuted, and the activation time courses were recomputed per feature. Then, the $d'$ statistic was calculated for each permuted time course. We used 10000 permutations per feature. The actual $d'$ values computed on the original non-permuted time courses were considered significant if they fell in either above the 99th (specificity to speech) or below the 1st (specificity to music) percentile of the surrogate distributions. These values are highlighted in **Fig 5B**.

## Acoustic patterns captured in key BO-NN features

A ridge linear regression was used to predict the spectromodulation features (STMFs) from the full set of the top-layer BO-NN features as well as the set of 53 key BO-NN features. Separate regression models were trained to predict STMFs during speech and music fragments. The regularization parameter was determined through the nested cross-validation (ten nested folds). The prediction accuracy scores were cross-validated across six sound-specific blocks. Significance testing was performed in the same way as described previously by creating a baseline distribution of prediction accuracies using shifted STMF time courses ($n = 1000$). Similar to the previous analyses, the significance threshold was set at the 99.999th percentile of the baseline distribution, which corresponds to a $p$-value of .001.

Feature-to-feature Spearman correlations between the key BO-NN features and the STMFs were computed separately for speech and music fragments. The data across all sound-specific blocks was concatenated into one vector and correlated between the feature sets. Significance testing was based on the correlations between the key BO-NN features and the shifted modulation feature time courses ($n = 1000$, shifts > 6 s). The $p$-value threshold was set at the 99.999th percentile of the baseline correlation distribution.

The STMF correlation maps over key BO-NN features (individual lines in **Fig 7B**) were created by selecting BO-NN features with significant correlation to STMFs and identifying the combination of values along the dimensions of SMs, TMs and frequency, associated with the highest correlation. The highest correlations were then plotted separately for each STMF dimension (SMs, TMs and frequency). The procedure was carried out separately for correlations during music and speech fragments.

To investigate the selectivity of each key BO-NN feature in tuning to a specific STMF dimension we examined the variances in correlation for each key BO-NN feature with the values along each STMF dimension separately. First, we identified the combination of values along dimensions of SMs, TMs and frequency, associated with the highest correlation with the given BO-NN feature. Then, we kept the STMF values of two dimensions fixed and only varied

the values along the third STMF dimension computing the correlation for each value with the given BO-NN feature. After that, we calculated the variance in BO-NN-STMF correlations along that STMF dimension. The procedure was repeated for each STMF dimension and the dimension of the largest correlation variance was marked for each key BO-NN feature. This would be the STMF dimension the given key BO-NN feature is selectively tuned to. The procedure was carried out separately for correlations during music and speech fragments.

For example, for the key BO-NN feature #2 we found that in speech its largest correlation is with a STMF combination of SM = 2.8 cyc/oct, TM = 4 Hz and frequency = 2.7 kHz. First, we fixed the values along the TM and frequency dimensions (to 4 and 2.7 respectively) and varied the values along the SM dimension, calculating the correlation between each combination of SM×4×2.7 with the BO-NN feature #2. This resulted in a vector of correlations over all SM values used in the study. Then, the procedure was repeated for the TM dimension (2.8×TM×2.7) and the frequency dimension (2.8×4×frequency). Then we computed the variance along each resulting vector of correlations: $\sigma^2_{SM} = 3 \times 10^{-4}$, $\sigma^2_{TM} = 0.18$ and $\sigma^2_{Freq} = 0.004$. This means that the changes in values along the SM and frequency dimensions affect the BO-NN-STMF coupling less than changes along the TM dimension, effectively making the key BO-NN feature #2 selectively tuned to the TM dimension and specifically to the TM value of 4 Hz.

The key features were then separated into groups depending on the STMF dimension they were selectively tuned to (the dimension of the highest variance in correlation) and shown in **Fig 7B** using a combination of a rug plot and a kernel estimation density plot per dimension and per sound type. The values of the Gaussian kernel density were estimated using the *statmodels* (www.statsmodels.org) Python library with default parameters.

## Temporal profiles of the key BO-NN features

Autocorrelations were calculated for each of the key BO-NN features using the entire feature time course. The autocorrelation value was determined as a time point as which the autocorrelation dropped below $r = .5$.

Latency profiles were calculated only for the speech fragments. First, we trained a ridge linear regression to predict the time courses of the key BO-NN features based on the simple speech audio characteristics, such as intensity, pitch, formant values and a binary regressor of speech being on or off (based on the manual annotation of the speech fragments). The continuous speech regressors (intensity, pitch and formants) were extracted automatically from the soundtrack using Praat software [50]. The binary regressor (speechON) was labeled manually by annotating the speech fragments with onsets and offsets of individual words. To construct the binary speech ON/OFF regressor, all the time points between consecutive onsets and offsets of the words were labeled as ones and remaining time points were labeled as zeros. The key BO-NN feature activations were then modeled as a linear combination of all the simple speech regressors. The details of training and cross-validation of the ridge linear regression were identical to the ones described above with the exception of additional training of separate models for different shifts between the speech regressors and key BO-NN feature activations to account for a possible temporal lag in the BO-NN feature activation to the speech regressor. A separate model was trained per each shift within -.5 to 1 second. The significance testing was performed in the same way as previously described by using shifted time courses and constructing a baseline distribution of the performance accuracy. In this case, the shifts were larger than 6 seconds. The latency per key BO-NN feature was determined as the maximal prediction accuracy, falling above the 99.999[th] percentile of the baseline correlation distribution, similarly to the previous analyses using the baseline shifted distribution to determine significance.

## Posterior-anterior gradient of temporal information in the perisylvian cortex during speech perception

In order to combine the results of various analyses aimed at interpretation of the key BO-NN features we constructed a matrix that per feature contained the following information: the correlation with STMFs (column 1); the prediction accuracy from the fit using speech Praat features (column 2); the mean normalized cortical $\beta$-weights (as a result of the linear regression from top BO-NN features to the HFB responses, column 3); the percentage of the electrodes with high cortical weights ($\beta$-weights $> 2$) in the perisylvian cortex (column 4); and the ratio of the electrodes with high weights in the perisylvian cortex to the weights in the rest of the brain (column 5). We considered the electrodes to belong in the perisylvian cortex if they were located in STG, precentral, postcentral gyri or IFG. Electrodes from precentral and postcentral gyri were only considered if they were in close proximity to the Sylvian fissure (z coordinate $< 10$).

Next, we used the *scikit-learn* implementation of the metric multi-dimensional scaling (MDS) algorithm [57] with default parameters to visualize and explore the resulting matrix (53×5) in a lower-dimensions space (number of dimensions = 3). To obtain the interpretation of each of the MDS dimensions we computed Spearman correlations between the MDS embedding values and each of the columns of the original matrix.

For the features that formed a cluster of highest values along the three MDS dimensions (**Fig 8C**), we quantified the posterior-anterior progression of the electrode locations associated with high cortical $\beta$-weights per feature. Per feature highlighted in **Fig 8C** we selected the electrodes with considerably large $\beta$-weights for that feature (zscored weights $> 2$). Of these, we only selected the electrodes along the perisylvian cortex (STG, precentral, postcentral gyri and IFG). Electrodes from precentral and postcentral gyri were only selected if they were in close proximity to the Sylvian fissure (z coordinate $< 10$). Because of the overall left hemisphere bias in coverage and the fact that most patients had left hemisphere as language-dominant, we also restricted the analysis to the left hemisphere electrodes. In total, 22, 18, 12, 36, 40, 26, 48, 46 and 67 electrodes entered the analysis for the key features # 41, #42, #17, #16, #49, #36, #1, #22 and #2, respectively (315 electrodes in total). The features were sorted according to their response latency profile from features with short lags with response to the audio onset (#41, #42) to the features with long lags with response to the audio onset (#22 and #2). An ordinary least squares fit was computed to find a linear relationship between the y-coordinate (anterior-posterior dimension of the electrode location) for the center of mass in the selected electrodes and the order of the associated key features. The slope of the fitted line would characterize the degree of the posterior-anterior anatomical progression (y-coordinate) over the key features from #41 to #2. The F(313,1) statistic comparing the model fit using the nine key feature labels against the constant model was computed. In addition, the highest correlation of each of the selected features along the TM dimension of STMFs was retrieved, and the associated TM values per ordered feature were reported.

## Acknowledgments

We thank Frans Leijten, Cyrille Ferrier, Geertjan Huiskamp, and Tineke Gebbink for help in collecting data; Peter Gosselaar and Peter van Rijen for implanting the electrodes; the technicians and staff of the clinical neurophysiology department and the patients for their time and effort; and the members of the UMC Utrecht ECoG research team for data collection. We also thank the BosBros film company and the Swedish Film Institute for their help and the provided materials.

## Author Contributions

**Conceptualization:** Julia Berezutskaya, Zachary V. Freudenburg, Nick F. Ramsey.

**Formal analysis:** Julia Berezutskaya.

**Investigation:** Julia Berezutskaya, Zachary V. Freudenburg, Nick F. Ramsey.

**Methodology:** Julia Berezutskaya, Umut Güçlü, Marcel A. J. van Gerven.

**Supervision:** Zachary V. Freudenburg, Umut Güçlü, Marcel A. J. van Gerven, Nick F. Ramsey.

**Writing – original draft:** Julia Berezutskaya, Nick F. Ramsey.

**Writing – review & editing:** Julia Berezutskaya, Zachary V. Freudenburg, Umut Güçlü, Marcel A. J. van Gerven, Nick F. Ramsey.

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
