## [Decision Letter · Decision Letter 0]

1 Nov 2019

Dear Dr Berezutskaya,

Thank you very much for submitting your manuscript 'Brain-optimized extraction of complex sound features that drive continuous auditory perception' for review by PLOS Computational Biology. Your manuscript has been fully evaluated by the PLOS Computational Biology editorial team and in this case also by independent peer reviewers. The reviewers appreciated the attention to an important problem, but raised some substantial concerns about the manuscript as it currently stands. While your manuscript cannot be accepted in its present form, we are willing to consider a revised version in which the issues raised by the reviewers have been adequately addressed. We cannot, of course, promise publication at that time.

Sincerely,

Frédéric E. Theunissen

Associate Editor

PLOS Computational Biology

Daniele Marinazzo

Deputy Editor

PLOS Computational Biology

[LINK]

Dear author,

As you see from the detailed comments of the three reviewers, your manuscript is not quite ready for publication. Some of the issues are simple request for more methodological details but I do see that these details might also change the interpretation of some of your results (for example the differences that you observe for music vs speech). I will thus carefully re-read your resubmission with this in mind. Also we believe that some of the processed data (without any subject identification) should be made publicly available and this is important for scientific rigor. Looking forward to your reply.

Best wishes,

Frederic Theunissen

Reviewer's Responses to Questions

**Comments to the Authors:**

Reviewer #1: In “Brain-optimized extraction of complex sound features that drive continuous auditory perception”, Berezutskaya et al. use a “brain optimized neural network” (BO-NN) to discover features driving neural activity recorded from ECoG during a movie watching task, and then use those features to predict neural data recorded from different participants on a separate natural listening task. They train a recurrent convolutional neural network to extract feature representations while 6 ECoG patients watch a Dutch movie. The model incorporates information from the raw audio waveform as well as 2D audio spectrogram features, and is optimized to predict responses to a full length feature movie with both auditory and visual information. The major novelty of this manuscript is in the ability to predict data from a completely separate cohort of participants listening to different movies. In particular, the authors show that their model captures responses to speech, but that responses to musical portions of the second stimulus set (Movie 2) were not as readily predictable. It is also very interesting that they are able to predict the responses in some higher order (presumably non-spectrotemporal) regions such as IFG, MTG, and postcentral gyrus. While I believe the results of this study will be interesting to readers of PLoS Computational Biology and I am enthusiastic about the use of a data-driven set of model features on an entirely new dataset, my main suggestions relate to interpretation of the neural network model features. There are a few details that I believe would aid in interpreting their results, especially with regard to comparing performance on speech vs. music tasks.

Major comments:

1. The differences in performance in music and speech are interesting, but I wonder whether these are driven by inherent differences in the acoustics of the training and test data. For example, in Figure 1, where only data from Movie 1 is used, the authors can predict responses to all classes of sounds, including music. However, once Movie 2 stimuli (alternating speech and music) are used, the performance of the model on musical stimuli decreases significantly. This leads me to a few questions:

a. Is there equal occurrence of speech and music in Movie 1 training set?

b. Are differences in speech and music predictions driven by similarity of speech stimuli from movie 1 and 2 and differences in acoustics for music in movie 1 and 2? An acoustic analysis (similarity of modulation spectra for each sound class in Movie 1 vs. 2) would be helpful here.

c. Could the authors comment on the music stimuli in Movie 1 vs. Movie 2 in terms of whether they use similar instruments, contain vocals or not, are similar genres, etc? Because music is a very broad category, it could be that results will not generalize to music unless music that was acoustically similar shows up in the training set (Movie 1).

2. Why was RSA used to compare the neural network models to the STG data? It would be more straightforward to compare the correlations directly for the random-NN, BO-CNN, and STMF models for the STG electrodes in the same way that this was done for non-STG electrodes. The correlations could be compared separately for music and speech and would be more easily interpreted than the RSA analysis.

3. Regarding the key features analyses, I think it is a great idea to try and correlate these with known quantities in an effort to better understand what the neural network has learned. However, I found this part of the manuscript to be the weakest part, since no specific suggestions or interpretations were made for each of the key features listed. I thus have a few specific questions:

a. Are key features 1, 2, and 36 (the speech specific features) actually tuned to any particular speech features? (e.g. certain spectrotemporal modulations, phonetic content, etc)

b. Key feature 35 is a strange one. Is it ever off, or is it constantly on in a sustained fashion (as described in the text)? It would make more sense if this was a “sound on” vs. “off” feature, but more description is needed.

c. Key feature 42’s cortical weight maps seem quite weak for music – is this real, or is this for a particular type of music (such as music with vocals?)

d. Key features 2 and 15 are described as being related to differences in speech vs. music – in particular, their activation time courses were less anticorrelated during music compared to speech. Could the authors provide some additional discussion related to this point? What exactly might it be about the speech vs. music stimuli that drives this?

4. I’m not sure I understand Figure 6b. How are the density plots related to the black lines? Are they simply fits to the data shown by the black lines? If so, is it the case that the BO-NN features only represent relatively high spectral modulations and no low spectral modulations? Are these low spectral modulations also not informative of the neural data? I’m surprised, as I would think that these, coupled with high temporal modulations, would predict some regions of posterior STG.

5. Similarly, I do not understand what Figure 7a is showing. Is this the time of the peak lag in the autocorrelation? If so, this doesn’t seem to completely match with the analysis in Figure 7b, which also shows the timescale of correlations with Praat features. How are the key features sorted? Please clarify.

6. With regard to Figure 7b, this figure would be much more helpful if it was clear which specific features (intensity, pitch, speech ON, or multiple) were contributing to each feature. Are they separable?

7. The authors should comment on how their mapping is or is not consistent with recent findings from other groups (e.g. Norman-Haignere et al. Neuron 2016, which showed posterior and anterior regions selective for non-vocal music that were separate from speech selective areas of the STG). This paper also showed some differential responses for music with and without singing, which could inform the question posed in #1 above.

Minor comments:

1. The statement in the abstract “The extracted bottom- up features captured acoustic properties that were specific to the type of sound and were associated with various response latency profiles and distinct cortical distributions” is quite vague. The authors should expand this to provide specific examples that they found from their results, or remove the sentence entirely.

2. In Figure 1, it could be helpful to connect the dots for each electrode in Figure 1b, since although the overall performance on speech vs. music, noise, and ambient sounds did not differ, it could be that those differences are observed in individual electrodes, with some tuned for different classes of sound.

3. The “transverse gyrus” is not standard nomenclature – it would be clearer to call this either the “transverse temporal gyrus” or “Heschl’s gyrus” or perhaps “transverse gyrus of Heschl”.

4. The authors note that the movie clips in Movie 2 were dubbed, so presumably the visual does not match the auditory stimulus. Perhaps they could comment on whether this influences any of the interpretations of their results.

5. The statement “code will be available upon reasonable author request” – I have seen this statement before, but authors were asked to change it to “upon request from the corresponding author”, as the word “reasonable” implies some value judgment on part of the authors, and the code should be made readily available. Data, on the other hand, could be made available through a University agreement so as to adequately protect patient related health information.

Reviewer #2: This study uses a neural network trained on ECoG patients watching a movie without strong theoretical assumptions to predict neural response from auditory stimulus. The study shows better prediction power compared to a specific model (spectrotemporal modulation) and analyzes the properties of the nodes in the final layer to shed light on the cortical properties of speech processing. Overall, it is an interesting study with some stimulating findings. For example, the set of properties that is represented by the network and sensitivity to spectrotemporal modulation, pitch, intensity, etc. is intriguing. On the other hand, given that the main conclusions of the paper rest on the properties of the representation in node activation of the network, more controls are needed to make the claims of the study more compelling.

Specifically:

1. It is not entirely clear why a multi-layer neural network is needed to show the claims of the study. The main point of the paper is the superiority of data-driven models over theory-driven models. For this argument, the data-driven model can be any regression framework, for example, STRF (linear), MID (Sharpee et. al.), or even a single-layer neural network. Given that all the analysis is done on the last layer of the network, why exactly the authors claim that multilayer hierarchical feature extraction is needed, and how the claims are dependent on this particular choice is unclear.

2. This particular network architecture (conv + RNN) was chosen because it had the best prediction accuracy. What is the effect of this model choice on the findings? E.g. do they not see the same acoustic features in the last layer of the random network (sensitivity to pitch, music, etc.) which can happen due to the random templates of the stimulus?

3. While it is interesting that the STM features are preserved in the neural network, it is perhaps more interesting to show what is added by the network that wasn’t present in theory-driven features. In other words, what is it that the network learns from the data that wasn’t already predictable from STM features? This information is important for the claim of the paper (advantage of data driven over theory)

4. The majority of the claims are based on single node response properties in the last layer of the neural network. While it is shown here that the population of nodes perform well in predicting the neural responses (linear projection of the last layer to the neural response), it is not clear whether the exact node properties are essential to the prediction, or whether they are preserved across different network implementations. In other word, different initialization of weights, different regularization of the network, different architecture (Conv, RNN, …), etc., can all result in the same input-output function and the information in the last layer (hence same prediction power from its linear projection), but the exact information that is represented in each node may not be the same and can change with changing network properties. Hence, the claim of the paper about the node properties is not very compelling unless the authors can show that these exact response selectivity by nodes is “necessary” for successful prediction of the neural data from the last layer, or at least consistently appear across models.

Reviewer #3: This manuscript is an interesting attempt to directly train a non-trivial DNN model to predict ECoG responses using about an hour of data from 6 patients collected during the viewing of a natural movie. The authors show the trained model outperforms a baseline spectrotemporal model (and an untrained model) in predicting responses to a shorter movie recorded in a different set of subjects (after linear re-mapping). The authors then take advantage of the fact that many more subjects were tested in the second movie in order to try and understand the organization of feature tuning in auditory cortex and higher-order regions using the model. The primary approach used is to cluster the learned features from the top-level of the trained model and to then investigate the selectivity of these features for different acoustic features or categories, and the extent to which different electrodes weight on these feature clusters.

Overall, I think there is promise in developing better training methods to directly predict neural responses and interpret the resulting features. At the moment, I am not sure there is a clear scientific conclusion that can be drawn from this work. But I think the approach is methodologically new enough that any progress along these lines is a useful contribution, with clearer scientific conclusions perhaps emerging down the road.

Focusing on the results highlighted in the abstract, the authors state “The extracted bottom-up features captured acoustic properties that were specific to the type of sound and were associated with various response latency profiles and distinct cortical distributions.” How could this be otherwise unless cortical selectivity was completely uniform?

They also state: “Our results also support and extend the current view on speech perception by demonstrating the presence of temporal hierarchies in perisylvian cortex and involvement of cortical sites outside of this region during audiovisual speech perception.” Figure 7, which is the data behind this claim, is interesting, and provides evidence that features with more delayed responses also have slower responses with stronger autocorrelations and coarser modulation tuning. I am not sure I find this result particularly surprising, but it’s a useful demonstration of how one can use the method. The supposed anatomical progression in Figue 7C is not particularly compelling in my opinion and is not quantified in any way.

I also found the discussion of audiovisual interactions potentially misleading since there will be correlations in the movie stimuli between auditory and visual features and thus one cannot conclude whether good predictions are driven by acoustic selectivity, visual selectivity or audiovisual selectivity.

Given that the primary contribution of this paper, in my opinion, is methodological (not a bad thing in my view), it would be nice for the authors to go into more detail about the various types of model architectures they tried, which types of model architectures worked best (and perhaps why), and any insights they gained along the way. Perhaps the authors could include a summary figure showing the performance of different models along with the spectrotemporal baseline, showing how different architectural choices led to improved performance. I think it might be useful to have a table that shows all of the architectural parameters for each layer of the final model (i.e. kernel size, stride, pooling size and dimension, activation function, etc.). The details should be sufficient such that someone else could easily implement the model. Upon publication, code should be made public on some type of repository, so that someone can download it freely (not upon “reasonable request” as is indicated).

More minor comments / points of clarification:

1. Figure 1a which schematizes the model I think could be clearer. My suggestions/questions: For the “spectra” input display a spectrogram so that it is clear what the input is and flip the axes so that time is on the x axis and frequency is on the y axis. For the “time” input put a waveform so that it is again clear what the input is (perhaps indicate that the filter number is 1 for both the spectra and waveform and the channel number is 1 for the waveform). For the lines/boxes that show the windows of convolution I think it would make sense to make the window size proportional to the filter size (i.e. 7x5 should be bigger than 1 x 1). Also, for the spectra stream, why are there blue cubes and colored planes? And why are there three planes/cubes when the filter number is 1? When the two streams are concatenated the time dimension appears to grow while the channel dimension appears to say fixed, which is the opposite of what occurs. In the last plane, it would be useful to indicate that 396 corresponds to the number of electrodes. How many recurrent layers are there? I would guess one recurrent layer followed by a 1x1 convolution layer to remix the outputs. But from the figure it is unclear. Also, when the authors say the “top layer”, I’m guessing they mean the one right before the final output used to predict the ECoG responses. It would be helpful to clarify this and perhaps make a distinction in the figure between the top layer and the final output.

2. Figures 4 & 5 are mostly descriptive, with little in the way of quantification, and there isn’t a clear result that one can draw from either. Maybe it would help to combine Figure 6 and Figure 4, and Figure 5 with Figure 7. Figures 5a and 5b don’t seem necessary to me (and plotting the max weight seems like an odd/un-robust statistic, which perhaps explains the noisy-looking pattern). Some quantification of anatomy seems important if the authors want to make anatomical claims. I don’t find the figures particularly compelling.

3. For Figure 7, why do the authors switch to using generic features from Praat, instead of the spectrotemporal features. If the authors are worried about delays in the spectrotemporal features, they could simple shift the filters so that they are centered/non-causal. Binarizing the autocorrelation function seems a bit strange to me. Why not just plot the ACF function and put dotted lines at the 50% mark? It would be nice to have a figure that quantifies the relationship between the autocorrelation width and delay, as well as the temporal modulation tuning and delay (a relationship suggested Figure 7c).

4. It was not clear to me how the RSA analysis was done. Typically one correlates the response pattern across the brain or across a model for pairs of stimuli. Then one correlates this stimulus distance matrix across models/brain regions. Here, they have a time-varying stimulus, so could in principle do this by comparing pairs of time-points, yielding a time-point x time-point matrix which can then be compared across models/regions. It seems like this is not what was done, and I couldn’t figure out exactly what was done and it how it relates to standard RSA.

5. In the first movie dataset, used for training, the authors show comparable prediction accuracy for different categories, but in the second movie, performance for music is worse than performance for speech. There is no explanation or discussion of this difference.

6. It would be nice if Figure 3b also showed prediction accuracy scores for STG. For this analysis how were the electrodes selected? Hopefully not based upon prediction accuracy of the DNN model as this would be biased?

7. For the spectrotemporal model, I think it would be useful to give the equations for the filters in the text. I know that this is published elsewhere, but it doesn’t take up much space and it’s useful to be sure about exactly what was done.

8. When examining speech/music selectivity it seems more sensible to use a measure like d-prime that incorporates the within vs. between category variance rather than “mean activation”.

9. Preference constant is not defined.

10. I assume that nested cross-validation means that there was cross-validation within the training set to pick the regularization parameters. It would be nice to spell this out and give the details (i.e. the number of folds for the inner and outer loop).

11. It would be nice to give statistics about the amount of speech, music and other categories present in the training movie.

12. The parametric t-stat computation shown in Figure 2 seems inappropriate since n presumably reflects the number of timepoints (not specified), but adjacent timepoints are presumably not independent.

13. The color-map in Figure 1 is hard to compare with the balls in the brain. Perhaps the authors could replace the color-bar with balls that have corresponding correlation values for easier comparison.

14. Were the electrode responses normalized or z-scored for any of the analyses?

15. There is no description or reference to the kernel density estimator used and how it was applied (i.e. Figure 6b). Also in Figure 6b, does each line correspond to the correlation value for the spectrotemporal feature with the highest correlation for a single “key feature”? If so, it would be good to clarify this in the legend.

16. Why did the authors use the beta weights to cluster the features as opposed to the similarity of the response timecourses? Perhaps there could be features with very different response selectivity but similar beta weights such that averaging these features into one cluster would be inappropriate?

**Have all data underlying the figures and results presented in the manuscript been provided?**

Reviewer #1: No: Data cannot be shared publicly due to restrictions on sharing patient data, but can be made available upon agreement with the University Medical Utrecht Center. It seems possible that deidentified data (or preprocessed data on model fits, for example) could be included without harm to patient identity.

Reviewer #2: Yes

Reviewer #3: No: No data have yet been provided.

PLOS authors have the option to publish the peer review history of their article (what does this mean?). If published, this will include your full peer review and any attached files.

Reviewer #1: No

Reviewer #2: No

Reviewer #3: No

---

## [Decision Letter · Decision Letter 1]

30 Apr 2020

Dear Ms Berezutskaya,

Thank you very much for submitting your manuscript "Brain-optimized extraction of complex sound features that drive continuous auditory perception" for consideration at PLOS Computational Biology. As with all papers reviewed by the journal, your manuscript was reviewed by members of the editorial board and by several independent reviewers. The reviewers appreciated the attention to an important topic. Based on the reviews, we are likely to accept this manuscript for publication, providing that you modify the manuscript according to the review recommendations.

Dear authors,

Please excuse the lateness of the re-review. One of your referees asked me specifically to tell you that he/she was sorry. I am sure that you understand. Please address the more minor comments raised (in particular by reviewer 3) in your next iteration.

Best wishes,

Frederic T.

Sincerely,

Frédéric E. Theunissen

Associate Editor

PLOS Computational Biology

Daniele Marinazzo

Deputy Editor

PLOS Computational Biology

[LINK]

Dear authors,

Please excuse the lateness of the re-review. One of your referees asked me specifically to tell you that he/she was sorry. I am sure that you understand. Please address the more minor comments raised (in particular by reviewer 3) in your next iteration.

Best wishes,

Frederic T.

Reviewer's Responses to Questions

**Comments to the Authors:**

Reviewer #1: In this revision of “Brain-optimized extraction of complex sound features that drive continuous auditory perception”, Berezutskaya et al. use a “brain optimized neural network” (BO-NN) to discover features driving neural activity recorded from ECoG during a movie watching task, and then use those features to predict neural data recorded from different participants on a separate natural listening task. Their main findings are that they can use features from a recurrent convolutional neural network to extract feature representations that can be used to predict neural data on a separate cohort of patients, and that there are some separable effects for speech and music in different anatomical areas. My previous review addressed a lack of interpretation of the “key features” from their models, and a need for direct comparisons between these and the spectrotemporal models. I believe this has been addressed nicely in the new revision of the manuscript. They have also included more details on the stimuli, including amount of speech, music, and other sounds (also requested by Reviewer #3). I agree with Reviewer #2 somewhat about point #3, which is that “it is interesting to show what is added by the network that wasn’t in the theory-driven features”. Still, I think the authors respond well by noting that interpretation of neural network models is often difficult, and I think they’ve done a good job at describing why the BO-NN might be better. Even if this paper doesn’t tell us exactly what is being represented by the NN, that is similar to many artificial neural network papers and is not unique to these authors, so I wouldn’t say the burden of proof is on them to show this. Still, I think it’s up to the reader (and future experiments) to determine what exactly is missing from our theories about what these areas “do” or “represent”!

I only have a few minor comments, which would not preclude acceptance but could be fixed in the final version:

New Figure 2:

• There is a small typo in the title, “Distritbution” should be “Distribution”.

• The semicircular plots are fine and aesthetically pretty, but I think the result would be clearer as a standard bar plot.

Reviewer #3: Uploaded as attachment.

**Have all data underlying the figures and results presented in the manuscript been provided?**

Reviewer #1: No: According to authors, data cannot be shared publicly due to confidentiality restrictions on sharing patient information, but will be made available from the University Medical Utrecht for researchers who meet criteria for access.

Reviewer #3: Yes

PLOS authors have the option to publish the peer review history of their article (what does this mean?). If published, this will include your full peer review and any attached files.

Reviewer #1: No

Reviewer #3: No
---

## [Editor Report · Decision Letter 2]

27 May 2020

Dear Ms Berezutskaya,

We are pleased to inform you that your manuscript 'Brain-optimized extraction of complex sound features that drive continuous auditory perception' has been provisionally accepted for publication in PLOS Computational Biology.

Best regards,

Frédéric E. Theunissen

Associate Editor

PLOS Computational Biology

Daniele Marinazzo

Deputy Editor

PLOS Computational Biology

---

## [Editor Report · Acceptance letter]

24 Jun 2020

PCOMPBIOL-D-19-01703R2 

Brain-optimized extraction of complex sound features that drive continuous auditory perception

Dear Dr Berezutskaya,

I am pleased to inform you that your manuscript has been formally accepted for publication in PLOS Computational Biology. Your manuscript is now with our production department and you will be notified of the publication date in due course.

With kind regards,

Laura Mallard
